



# Ice nucleation efficiency of natural dust samples in the immersion mode

Lukas Kaufmann[1], Claudia Marcolli[1,2], Julian Hofer[1,3], Valeria Pinti[1], Christopher R. Hoyle[4,5], and Thomas Peter[1]

[1]Institute for Atmospheric and Climate Science, ETH, Zurich, Switzerland
[2]Marcolli Chemistry and Physics Consulting GmbH, Zurich, Switzerland
[3]Leibniz Institute for Tropospheric Research (TROPOS), 04318 Leipzig, Germany
[4]Laboratory of Atmospheric Chemistry, Paul Scherrer Institute, Villigen, Switzerland
[5]WSL Institute for Snow and Avalanche Research SLF Davos, Switzerland

*Correspondence to:* C. Marcolli (claudia.marcolli@env.ethz.ch)

**Abstract.** Twelve natural dust samples from eight dust source regions on four continents were investigated with respect to their ice nucleation activity, revealing no significant differences between source regions. Dust collection sites were distributed across Africa, South America, the Middle East and Antarctica. Mineralogical compositions have been determined by means of X-ray diffraction. All samples proved to be mixtures of minerals, with major contributions from quartz, calcite, clay minerals, K-feldspars and (Na, Ca)-feldspars. Reference samples of these minerals were investigated with the same methods as the natural dust samples. Furthermore, Arizona Test Dust (ATD) was re-evaluated as a benchmark. Immersion freezing of emulsion and bulk samples was investigated by differential scanning calorimetry. For emulsion measurements, water droplets with a diameter of about 2 µm, containing different amounts of dust between 0.5 wt% and 50 wt% were cooled until all droplets were frozen. These measurements characterize the average freezing behaviour of particles, as they are sensitive to the average active sites present in a dust sample. In addition, bulk measurements were conducted with one single 1 mm diameter droplet consisting of a 5 wt% aqueous suspension of the dusts/minerals. These measurements allow the investigation of the best particles/sites available in a dust. All natural dusts except for the Antarctica and ATD samples froze in a remarkably narrow temperature range with the heterogeneously frozen fraction reaching 10% between 244 and 250 K, 25% between 242 and 246 K, and 50% between 239 and 244 K. Bulk freezing occurred between 255 and 265 K. In contrast to the natural dusts, the reference minerals reveal ice nucleation temperatures with 2 – 3 times larger scatter. Calcite, dolomite, dolostone and muscovite can be considered inactive as ice nuclei (IN). For microcline samples 50% heterogeneously frozen fraction occurred above 245 K for all tested suspension concentrations, and a microcline mineral showed bulk freezing temperatures even above 270 K. This makes microcline ($KAlSi_3O_8$) an exceptionally good IN, superior to all other analysed K-feldspars, (Na, Ca)-feldspars and the clay minerals. However, microcline is not abundant in the investigated natural dust samples. While K-feldspars were identified in five of the seven investigated natural source regions, only one sample contained microcline, and then only as a minor fraction. In summary, the mineralogical composition can explain the observed freezing behaviour of five of the investigated 12 natural dust samples, and partly for six samples, leaving the freezing efficiency of only one sample



not easily explained in terms of its mineral reference components. While this suggests that mineralogical composition is a major determinant of ice nucleation ability, in practice most natural samples consist of a mixture of minerals, and this mixture seems to lead to remarkably similar ice nucleation abilities, regardless of their exact composition, so that global models, in a first approximation, may represent mineral dust as a single species with respect to ice nucleation activity. However, more

sophisticated representations of ice nucleation by mineral dusts should rely on the mineralogical composition based on a source scheme of dust emissions.

## 1 Introduction

Freezing of droplets consisting of liquid water or aqueous solutions and subsequent ice crystal growth affects microphysical and radiative properties of clouds and precipitation. Understanding and predicting the formation of ice in clouds is critical to

quantifying the impact of aerosols on climate (DeMott et al., 2011). Ice crystals in the atmosphere may form by homogeneous ice nucleation of cloud droplets at temperatures $T < 237$ K or by heterogeneous ice nucleation under warmer conditions and possibly in the whole temperature range $T = 237 – 273$ K. For heterogeneous nucleation so-called ice nuclei (IN), i.e. particles that initiate a phase transition in an environment supersaturated or supercooled with respect to ice, need to come in contact with (contact freezing), be immersed in (immersion freezing), act as condensation nuclei for water droplets (condensation

freezing), or present a surface for deposition from the gas phase (deposition nucleation) (Pruppacher and Klett, 1997). Field measurements have shown that heterogeneous ice nucleation is indeed an important process in the atmosphere since ice formation in cumulus and stratiform clouds begins at temperatures much warmer than those associated with homogeneous ice nucleation in pure water (Korolev et al., 2003; Ansmann et al., 2009; Seifert al., 2010; Kanitz et al., 2011). Recently, Cziczo et al. (2013) suggested that heterogeneous ice nucleation might even have a dominating role in the formation of high, cold

cirrus clouds.

Ice nuclei are generally rare and may represent less than 1 in $10^6$ of the aerosol particle population (DeMott et al., 2011). Various insoluble particles such as mineral dust, soot, metallic particles, volcanic ash, or primary biological particles have been suggested as IN (Hoyle et al., 2011; Hoose and Möhler, 2012; Marcolli, 2014). The ability of mineral dusts to nucleate ice has been demonstrated in many laboratory experiments (Hoose and Möhler, 2012; and Marcolli, 2014; and references

cited therein), and their impact on cloud properties has been shown in observational and modelling studies (Lohmann and Diehl, 2006; Hoose et al., 2010; Choi et al. 2010; Ansmann et al., 2008; 2009; Seifert et al., 2010). In mixed-phase clouds mineral dust is usually the dominant IN (Pratt et al., 2009; Kamphus et al., 2010) and was also found to be important for the heterogeneous freezing of cirrus clouds (DeMott et al., 2003; Twohy and Poellot, 2005; Cziczo et al. 2013).

The main sources of mineral dusts in the atmosphere are the arid regions in the world (Prospero et al., 2002; Ginoux et al.

2012). Model estimates indicate that global mineral dust emissions by wind-driven erosion range between 1000 and 3000 Tg yr$^{-1}$ (Zender et al., 2004; Cakmur et al., 2006; Textor et al., 2007). The main source is the global dust belt, which stretches



from the Sahara to the Taklimakan in China. Sources outside the global dust belt are deserts located in the USA and Mexico, Australia, Botswana and Namibia, Bolivia, and in western Argentina (Sassen et al., 2003; Engelstaedter et al., 2006; Formenti et al., 2011). From these source regions the dust may be transported in a wide altitude range over large distances (Wiacek and Peter, 2009; Wiacek et al., 2010; Engelstaedter et al., 2006; Li et al., 2008; Reid et al., 2003).

Immersion freezing is often considered the most important nucleation mode for mineral dusts (Hoose et al., 2010; Murray et al., 2012; Wiacek et al., 2010). The ability of mineral dusts to act as IN depends on their mineralogical composition (Murray et al., 2012; Atkinson et al., 2013), but may also be influenced by the presence of coatings or biological material (Baker et al., 2005; Möhler et al., 2008; Cziczo et al., 2009; Pratt et al., 2009; Sullivan et al., 2010a; Conen et al. 2011; Hallar et al., 2011). The surface structure and therefore the nucleation ability of mineral dust can change due to interaction with organic or
inorganic substances. The effect of a coating depends on many different factors such as the mode of freezing, freezing temperature, and thickness and chemical composition of the coating (Cziczo et al. 2009; Chernoff and Bertram 2010; Sullivan et al., 2010a,b; Tobo et al., 2012). Clay minerals and Arizona Test Dust (ATD) show a decrease of the nucleation ability due to coatings (Cziczo et al. 2009; Chernoff and Bertram 2010; Sullivan et al. 2010a). Coatings that are acquired by dust particles during transport might be of minor importance for ice nucleation once a liquid cloud has formed and freezing occurs in
immersion mode. In the absence of surface chemical reactions, the coating may dissolve in the cloud droplet so that the bare surface is exposed again and the ice nucleating ability is restored (Sullivan et al., 2010b; Tobo et al., 2012; Kulkarni et al., 2014; Wex et al., 2014). Internal mixing of mineral dust particles with biological material is supposed to increase the nucleation ability of the dust and might occur during cotransport or by lifting of soil particles (DeMott et al., 2003; Baker et al., 2005; Pratt et al., 2009; Conen et al., 2011; Hallar et al., 2011; Creamean et al., 2013).

The most important components of mineral dusts are typically quartz, clay minerals, feldspars, and calcite (Murray et al., 2012). Clay minerals such as kaolinite, montmorillonite or illite have usually been considered the main responsible components in mineral dusts for ice nucleation. Early studies indeed found kaolinite and montmorillonite among other minerals, as inclusions of atmospheric ice crystals (Kumai, 1961; Kumai and Francis 1962). Many studies used therefore clay minerals as surrogates for mineral dusts (Hoose and Möhler, 2012; Welti et al., 2014; Wex et al., 2014; Wheeler et al., 2015;
Augustin-Bauditz et al., 2014; Marcolli, 2014; Hiranuma et al., 2015; and citations therein). Recently, Atkinson et al. (2013) suggested that feldspar particles may account for a large proportion of IN that contribute to freezing just below about 258 K.

To investigate the role of mineral dust as an IN, and the dependence of ice nucleation activity on mineralogical composition, natural dust samples from global dust source regions must be analysed (e.g. Pinti et al. 2012). Dust collected directly from the air as well as dust collected from the ground in regions, where dust events occur frequently, should be compared with mineral
reference samples in order to give new insights into the influence of mineral dusts on freezing in mixed-phase clouds. This study presents freezing results for natural dust samples collected from the ground at reported source regions of atmospheric mineral dust around the globe. To preserve the morphology, the only processing applied to the samples was sieving. We investigated whether the ice nucleation efficiency of the samples show significant differences between different source





regions and whether the freezing behaviour can be related to the mineralogical composition. To this end, a set of reference minerals was subjected to freezing experiments using the same procedure as for the natural dust samples.

## 2 Experimental setup

For the immersion freezing measurements we used a differential scanning calorimeter (DSC) Q10 from TA instruments. We performed both emulsion freezing and bulk freezing experiments, the former to characterize the average nucleation efficiency of dust particles, the latter to investigate the best available particles/sites in a dust.

First, for emulsion freezing experiments, a quantity of sieved dust, or a powder of the mineral in the case of the reference samples, was mixed with "Water Molecular Biology Reagent" from Sigma-Aldrich, which proved to have lower average freezing temperatures of droplets compared with our Milli-Q water. Then, 20 vol% of this suspension plus 80 vol% of a mixture of 95 wt% mineral oil from Aldrich Chemical and 5 wt% lanolin from Fluka Chemical were emulsified with a rotor-stator homogenizer (Polytron PT 1300D with a PT-DA 1307/2EC dispersing aggregate) during 40 seconds at 7000 rpm. Next, 4 – 15 mg of this emulsion was placed in an aluminium pan, hermetically sealed, and subjected to three freezing cycles following the method developed by Marcolli et al. (2007). The first and the third freezing cycles were executed at a cooling rate of 10 K/min to control the stability of the sample. The second freezing cycle was executed at a 1 K/min cooling rate and was used for evaluation. The evaluation was done using the implemented software "TA Universal Analysis" of the instrument. The heterogeneous freezing peak was analysed in terms of three characteristic temperatures that correspond to 10% ($T_{het,10\%}$), 25% ($T_{het,25\%}$), and 50% ($T_{het,50\%}$) of heterogeneously frozen water volume. This value was obtained by integrating over the heat flow signal of the heterogeneous freezing peak and setting the total heterogeneous heat flow to 100%. If the homogeneous and heterogeneous peaks overlapped, the heterogeneous peak was assumed to end where the heterogeneous peak shape was clearly influenced by the homogeneous peak (see Appendix A for further information). The integrals over the heterogeneous and homogeneous peaks are used to calculate the ratio between homogeneous and heterogeneous freezing. Emulsion measurements were performed with 0.5 wt%, 1 wt%, 2 wt%, 5 wt%, and 10 wt% mineral dust suspensions. For calcite, dolomite, dolostone and ankerite it was hardly possible to evaluate the 10%, 25% and 50% heterogeneously frozen water volume because of a very weak heterogeneous freezing signal. Due to this, these minerals were also measured with 50 wt% suspension concentration, and onset temperatures were evaluated. The evaluation method for onset temperatures is described in Zobrist et al. (2006). Very abrupt, but small spikes are excluded from the evaluation, because they originate from single droplets in the tail of the size distribution, which are orders of magnitude larger in volume than the average droplet. For dust samples with many large particles (from micrometers upwards), spikes were more frequent. Comparison of the evaluated temperatures for different samples of the same dust with the same concentration usually showed standard deviations of 0.5 K. Samples with weak heterogeneous signal or with spikes overlapping the smooth heterogeneous signal developed standard deviations up to 1 K.





Second, for bulk freezing experiments the investigated dusts and minerals were mixed with pure water. Of this suspension, 1.8 – 2 mg were placed in an aluminium pan, covered with mineral oil to avoid evaporation or condensation and finally hermetically sealed. The sample was subjected to repeated freezing cycles with 10 K/min, which is a suitable cooling rate as the freezing temperature is given by a clear heat release onset due to the sudden freezing of the whole droplet. Bulk measurements were performed with a 5 wt% suspension. Blank bulk freezing experiments with the pure water showed freezing usually at approximately 250 K. The highest freezing temperature of pure water was observed at 252.5 K. Freezing at higher temperatures is therefore attributed to the presence of the dusts and minerals in the samples.

Pictures for the evaluation of the size distribution of emulsion droplets were taken with an optical microscope (Olympus BX-40). The droplet diameter was evaluated with the free image processing and analysis program "ImageTool" (from the University of Texas Health Science Center at San Antonio), grouped into bins with a width of 0.5 µm from 0 µm to 10 µm, and fitted with a lognormal number distribution

$$N(d) = \frac{N_0 e^{w^2/2}}{\sqrt{2\pi} w d} e^{-\frac{\ln^2(d/d_m)}{2w^2}}, \tag{1}$$

giving $d_m = (2.41 \pm 0.04)$ µm as the droplet mode diameter, and $w = 0.507 \pm 0.014$ as the mode width, where $N_0$ is the total number of droplets. The volume distribution was fitted with the function

$$V(r) = \frac{V_0 e^{w^2/2}}{\sqrt{2\pi} w d} e^{-\frac{\ln^2(d/d_m)}{2w^2}}, \tag{2}$$

where $d_m = (6.1 \pm 0.4)$ µm, $w = 0.53 \pm 0.05$, and $V_0$ is the total volume of all droplets. The obtained number and volume distributions are shown in Fig. 1.

Size distributions of dusts were measured with a TSI 3080 scanning mobility particle sizer (SMPS) and with a TSI 3321 aerodynamic particle sizer (APS) and combined as described by Beddows et al. (2010). All natural dust samples were sieved with a 32 µm sieve. No other treatment was applied, except for the Antarctic dust where milled samples were used in addition to the untreated sample. Similar to the size distributions of the emulsion droplets, also the dust number size distributions were fitted to a lognormal distribution. Results are given in Table 1. Reference minerals were provided by the Institute of Geochemistry and Petrology of ETH Zurich and milled with a tungsten carbide ball mill. No additional treatment was applied. The mineralogical composition was measured by X-ray diffraction (XRD) of the powder samples. A quantitative analysis was performed with the "AutoQuan" program which is a commercial product of GE inspection technologies and which makes a Rietveld refinement (Rietveld, 1967; 1969).



## 3 Statistical evaluation of emulsion measurements

Knowing the droplet size distribution of the emulsions and the size distribution of the dust particles, the theoretical value for the latent heat release for homogeneous and heterogeneous freezing can be estimated. This theoretical value can be directly compared with the measured latent heat release by the DSC.

To calculate the total number of dust/mineral particles in an emulsion, the average volume of particles $\bar{V}_p$ was calculated by

$$\bar{V}_p = \int_0^\infty N(r) \frac{4}{3} \pi r^3 \, dr \quad , \tag{3}$$

where $N(r)$ is the normalized particle size distribution of the respective mineral/dust as a function of its radius $r$. With the mass of the dust, $m_{dust}$ and assuming a density of dust, $\rho_{dust} = 2.6$ g/cm$^3$, the number of dust particles $n$ in an emulsion can be estimated to be

$$n = \frac{m_{\text{dust sample}}}{\rho_{\text{dust}} \bar{V}_p} \quad . \tag{4}$$

The probability $P_j$ for a particle to be in a droplet $j$ with a volume $V_j$ is $P_j = V_j/V_{tot}$, where $V_{tot}$ is the total volume of all droplets in the emulsion. Assuming $n$ particles in the emulsion, which are all distributed among the water droplets, the probability for no particle in a droplet $j$ with a volume $V_j$ is $(1-V_j/V_{tot})^n$. The contribution of droplet $j$ to the total heterogeneous and homogeneous peak area $A_{tot}$ is proportional to $V_j/V_{tot}$. The percentage of homogeneous freezing, $p_{hom}$, can then be written as

$$p_{hom} = \sum_{j=1}^k \frac{V_j}{V_{tot}} \cdot \left(1 - \frac{V_j}{V_{tot}}\right)^n , \tag{5}$$

where $k$ is the number of droplets. The fraction of heterogeneously frozen volume $p_{het}$ is then

$$p_{het} = 1 - p_{hom}. \tag{6}$$

Like this, we are able to calculate the heterogeneously frozen water volume under the assumption that all dust/mineral particles are able to induce heterogeneous freezing.

The fraction of heterogeneously frozen volume measured with the DSC ($p_{het,lab}$) is calculated by dividing the latent heat release of the heterogeneous freezing signal by the latent heat release of the total freezing signal (homogeneous plus heterogeneous). The number of dust/mineral particles, $n_{lab}$, necessary to explain the heterogeneously frozen volume fraction can be obtained by solving the equation

$$p_{het,lab} = 1 - \sum_{j=1}^k \frac{V_j}{V_{tot}} \cdot \left(1 - \frac{V_j}{V_{tot}}\right)^{n_{lab}} \tag{7}$$

for $n_{lab}$. The active fraction $f_{act}$ of dust or mineral particles, i.e. the fraction of particles that are active as IN, can then be calculated by

$$f_{act} = \frac{n_{lab}}{n} \quad . \tag{8}$$





Correspondingly, $1 - f_{\mathrm{act}}$ is the fraction of dust (or mineral) particles that remain inactive to the point that droplets freeze homogeneously. By calculating the heterogeneously frozen fraction $p_{\mathrm{het,lab}}$ as a sum of droplet volume bins, it is taken into account that freezing of a larger droplet contributes more to the latent heat signal than freezing of a smaller droplet. The probability for a droplet to contain at least one particle depends on its volume. With increasing volume the probability

increases that a droplet contains at least one particle. Therefore, the probability for a larger water droplet to freeze heterogeneously is higher than for a smaller water droplet. Hence, freezing of larger droplets dominates the heterogeneous freezing signal and freezing of smaller droplets the homogeneous freezing signal. For example, assuming the latent heat release of the heterogeneous freezing signals to be the same as the latent heat release of the homogeneous freezing signal, the heterogeneous freezing signal represents the freezing of fewer larger droplets and the homogeneous freezing signal the

freezing of a high number of smaller droplets.

## 4 Minerals and dust samples

### 4.1 Dust samples

Natural dust samples have been collected from Antarctica, Bolivia, Etosha (Namibia), Israel, Makgadikgadi (Botswana), Oman and Qatar. For comparison, commercially available Arizona test dust (ATD) was also investigated. The geographical

locations of all eight sampling sites are shown in Fig. 2. The natural dusts are briefly characterized below.

(1) **The Antarctica sample** was collected at 74°16.5' south and 9°37.3' west at an altitude of 1520 meters above sea level. Antarctica is generally not considered as a source region for atmospheric dust, rather, dusts from dominant dust source regions in the southern hemisphere reach the Antarctica (Revel-Rolland et al., 2006; Winckler et al., 2008; Li et al., 2008; Genthon, 1992). Nevertheless, the fine (sieved) fraction of the Antarctica sample is not transported dust but from

local erosion, because it exhibited a very similar XRD pattern as larger grains or gravels from the same sample that are too large to be transported. To see the influence of milling, the fine fraction (particles $< 32$ µm) of the Antarctica sample was also milled.

(2) **Arizona test dust (ATD)** is a commercial dust sample that has been used by many investigators as a proxy of natural atmospheric mineral dust (Murray et al., 2012). It is produced by grinding samples of sand from Arizona. It is described

in detail by Möhler et al. (2006) and Knopf and Koop (2006).

(3) **The Bolivia sample** was collected in the surroundings of the Laguna Verde in the southern part of Bolivia. The salt flats on the Altiplano including Laguna Verde are important dust sources in South America (Goudie and Wells, 1995; Washington et al., 2003).

(4) **The Etosha sample** was collected in the Etosha pan in the northern region of Namibia. The pan was formed by a lake

which dried out. Bryant et al. (2007), Prospero et al. (2002) and Washington et al. (2003) describe the Etosha pan as one



of the principle dust sources in southern Africa. The Etosha sample was collected 18.0143° south and 16.0118° east at the eastern part of the pan.

(5) **Hoggar Mountain dust** was collected from the Sahara region (Pinti et al., 2012). It is a mixture of minerals originating from a source region with high shares of clay minerals (Pinti et al., 2012). According to Laurent et al. (2010) the Hoggar Mountains are part of a region which has been identified as a major source for desert dust aerosols.

(6) **The Israel samples** were collected close to Sde Boker in a dried out river bed. Poor vegetation is present in this region. The region itself is frequently exposed to desert dust due to its position at the northern end of the Negev desert, but is not itself believed to be a supra-regional atmospheric dust source (Prospero et al., 2002).

(7) **The Makgadikgadi samples** were collected in the Makgadikgadi pans in the northeast of Botswana. In the past there were lakes which dried out. Prospero et al. (2002) and Bryant et al. (2007) describe Makgadikgadi pans as an important source for atmospheric mineral dust. Material was collected from three different places and analysed separately as sample numbers A, B, and C. Makgadikgadi A was collected at 20.60°S and 25.22°E directly in the Ntwetwe pan. There was a light crust on the surface, which is probably salt, in accordance with the halite component detected in the XRD measurements. Makgadikgadi B was collected at 20.71°S and 25.21°E in the Ntwetwe pan, too. There was again a light crust on the surface and halite was also found in the XRD measurements. Makgadikgadi C was collected at 21.01°S and 25.06°E at a southern branch of the Ntwetwe pan. The sample was collected next to a fence, where cattle were present. There was no crust.

(8) **The Oman dune sample** was directly collected in the desert from dunes at 25.0995°N and 51.34°E. Dunes consist mainly of coarse material, what is also reflected by the fact, that a large amount of dune sand was necessary to gain small quantities of material consisting of particles smaller than 32 μm in diameter. Nevertheless Prospero et al. (2002) show strong dust events in this region.

(9) **The Qatar dune sample** was directly collected in the desert from dunes at 25.10°N and 51.34°E. This is also a region with strong dust events (Prospero et al., 2002).

## 4.2 Minerals

To correlate the freezing behaviour of natural dust samples with their mineralogical composition, reference minerals provided by the Institute of Geochemistry and Petrology of ETH Zurich were milled and investigated the same way as the natural dust samples.



## 5 Results

### 5.1 Natural dust samples

#### 5.1.1 Freezing experiments

Figures 3a and 3b show the thermograms of the emulsion measurements of all natural dust samples for suspension

concentrations ranging from 0.1 – 10 wt%. The investigated natural dusts exhibit a homogeneous freezing peak with maximum at 235.0 – 235.5 K and heterogeneous freezing in the wide temperature range of 236 K – 252 K. In the thermograms of the samples from Antarctica, Bolivia, Etosha and Makgadikgadi A, two or even three heterogeneous freezing peaks appear. The intensity of these peaks varies depending on the suspension concentration. Peaks at low temperatures prevail for low concentrations while with increasing concentration peaks at higher temperatures gain intensity because the

number of particles per droplet increases. The freezing behaviour of a droplet including many IN is controlled by the most potent one, which nucleates ice at the highest temperature. Therefore, less potent IN are dominated by the most potent ones at high concentrations. Table 2 shows the active particle fraction ($f_{act}$) together with $p_{het}$ and $p_{het,lab}$, the calculated and measured heterogeneously frozen water volume fractions. These quantities have been determined according to Eqs. (6) – (8). In addition, Table 2 gives the droplet diameter with an average of one particle inside ($D_{p1}$) for 2 wt% suspension concentration,

which is lowest for the Bolivia (2.0 µm) and highest for the Etosha sample (5.5 µm). The calculated number of dust particles per droplet depends on the measured number size distributions shown in Table 1, which peak between 216 nm (Bolivia sample) and 479 nm (sample Israel 2). For all dust samples $p_{het} > p_{het,lab}$, implying that only a fraction of the particles present in the natural dusts are active as IN. The active particle fraction $f_{act}$ is lowest for the Bolivia sample (0.025) and highest for the Etosha sample (0.32). There are considerable uncertainties associated with $f_{act}$ that are discussed in detail in Appendix A.

Based on this uncertainty evaluation $f_{act}$ may be up to 5.8 times larger and 2.3 times smaller than the values given in Table 2. Due to the fact that not enough material of the Oman and Qatar sample was available for particle size distribution measurements, no evaluation was possible for these samples.

A summary of DSC measurements of all dusts is given in Fig. 4 together with results for the reference samples, which will be discussed later. Figure 4 displays the evaluation of the heterogeneous freezing peak temperatures of the natural dust samples

with respect to 10% ($T_{het,10\%}$), 25% ($T_{het,25\%}$), and 50% ($T_{het,50\%}$) water volume frozen heterogeneously. This method of representation was chosen to obtain characteristic heterogeneous freezing temperatures to compare the different samples. For all samples, an increase of the characteristic temperatures with increasing suspension concentration can be observed. Blue symbols show onsets of heterogeneous freezing for bulk samples. With each sample a series of freezing cycles were run which are all designated by the same symbol in Fig. 4. Bulk experiments run with different portions of the same stock

suspensions are represented by different symbols.

The natural dust samples all show surprisingly similar freezing temperatures for emulsion freezing, with the notable exceptions of Antarctica and ATD. Indeed, 10% heterogeneously frozen fraction for natural dust samples (except Antarctica



and ATD) is realized between 244 and 250 K. The 25% heterogeneously frozen fraction is between 242 and 246 K, and the 50% heterogeneously frozen fraction is between 239 and 244 K. Also bulk freezing (with 5 wt% suspension concentration) occurs in a compact temperature range between 255 and 262 K (except for ATD, the Antarctic sample and one sample from Bolivia).

5   For ATD in emulsions, the 10% heterogeneous frozen fraction is at 250.5 K, the 25% heterogeneous freezing fraction is at 249.5 K and the 50% heterogeneously frozen fraction is at 248 K. Bulk freezing occurs at comparatively high temperatures between 262 and 266 K. For the Antarctica sample the 10% heterogeneously frozen fraction is between 249 and 252 K, 25% between 245 and 248 K and 50% between 243 and 245 K. Bulk freezing occurs between 261 and 264 K.

### 5.1.2 Mineralogical composition

10   Table 3 lists the most frequent minerals identified in the natural dust samples by the Rietveld refinement of the X-ray diffractograms. We consider the mineralogical composition determined by XRD diffraction as accurate within ± 15% (see Appendix B). Minor components might remain undetected.

The Antarctica sample has high shares of K-feldspars and (Na, Ca)-feldspars, but contains also muscovite and quartz. The sample from Bolivia has high shares of clay minerals (kaolinite, smectite), but contains in addition calcite ($CaCO_3$), quartz, 15 and plagioclase. The Etosha sample mainly consists of carbonates, namely calcite ($CaCO_3$), dolomite ($CaMg(CO_3)_2$), and ankerite ($Ca(Fe,Mg,Mn)(CO_3)_2$). The samples from Israel contain calcite as main component with over 60% together with minor fractions of quartz, ankerite, the clay minerals illite and smectite (montmorillonite), muscovite (mica), and the feldspars sanidine and plagioclase. Samples from Makgadikgadi show quite diverse mineralogical compositions containing calcite, muscovite, quartz and clay minerals but hardly any feldspars. The dune samples from Oman and Qatar mainly consist of 20 quartz, calcite, and the (Na, Ca)-feldspar plagioclase. Finally, the sample from Qatar also contains dolomite and the K-feldspar microcline, the one of Oman the K-feldspar sanidine.

### 5.2 Reference minerals

The analysed reference minerals can be classified in four groups, as shown in Table 1. The first group consists of the most frequently found minerals in the natural dust samples, namely quartz, muscovite, and the carbonates calcite, 25 dolomite/dolostone and ankerite. The second group are K-feldspars (adularia, microcline, orthoclase and sanidine), which are supposedly very efficient IN (Atkinson et al., 2013). The third group consists of (Na-Ca)-feldspars (anorthite, labradorite, pericline and plagioclase), which have been reported to be quite efficient IN (Atkinson et al., 2013; Augustin-Bauditz et al., 2014), though less effective than K-feldspars. The fourth group are clay minerals: DSC freezing experiments of illite, kaolinite and montmorillonite (member of the smectite group) have been performed by Pinti et al. (2012) and can be used here 30 as references for clay minerals. Most of the other minerals that could be identified in the natural dust samples, are water soluble and therefore not relevant for immersion freezing.





### 5.2.1 Mineralogical composition

Table 4 lists the mineralogical composition of the reference samples identified by the Rietveld refinement of the X-ray diffractograms. In most cases, the Rietveld analysis yields compositions in agreement with the identification of the stones by the Institute of Geochemistry and Petrology of ETH Zurich (see Sect. 2). Mineralogically pure or almost pure samples of the assigned composition proved to be calcite, dolomite, muscovite, quartz, sanidine, and plagioclase. However, for the K-feldspars adularia and orthoclase, the composition assigned by X-ray diffraction was different from the geological identification of stones. The adularia stone proved to be 100% sanidine, and the orthoclase stone consisted mostly of sanidine (77%) with no share of orthoclase. Sanidine is the high temperature polymorph of orthoclase, i.e. also a K-feldspar. We therefore obtained a second stone of adularia and of orthoclase, which we labelled adularia 2 and orthoclase 2. Indeed, adularia 2 proved to be 100% adularia, however, orthoclase 2 contained 78% of the K-feldspar microcline with no share of orthoclase. Therefore, we do not have a reference for orthoclase. Two stones of ankerite ($Ca(Fe,Mg,Mn)(CO_3)_2$, trigonal) were tested. For the first stone, the Rietveld refinement yielded 99.5% calcite ($CaCO_3$, trigonal) and only 0.5% ankerite, the second one consisted only of calcite with no ankerite present at all. The mineralogical similarity between ankerite and calcite might be the reason why ankerite was mistaken as calcite. Dolostone is a sedimentary carbonate rock that contains a high percentage of the mineral dolomite ($CaMg(CO_3)_2$). Rietveld analysis of the X-ray diffractogram identified it as 100% dolomite. The microcline stone from Elba (microcline E) contained 90% microcline and the one from Namibia (microcline N) 77% microcline, both with plagioclase as minor component. Albite (pericline), anorthite, plagioclase, and labradorite are all members of the plagioclase (Na, Ca)-feldspar solid solution series with triclinic crystal symmetry. All of them were identified as plagioclase by our XRD analysis. The plagioclase and labradorite samples were identified as pure plagioclase, for albite (pericline) minor shares of quartz (16%) were also found. For the anorthite sample in addition to plagioclase as the main component (59%), 12% quartz, 8% muscovite and 5% kaolinite were identified as most abundant minor components. It is not clear whether the Rietveld refinement was not able to discriminate between the different (Na, Ca)-feldspars or whether indeed all samples consisted of plagioclase.

### 5.2.2 Freezing experiments

DSC thermograms of the reference minerals are shown in Figs. 5a-c for concentrations ranging from 0.5 to 10 wt%. A summary plot is given in Fig. 4. Table 5 shows the active particle fraction $f_{act}$. For the samples of calcite, dolomite, dolostone and ankerite, it was hardly possible to evaluate the 10%, 25% and 50% heterogeneously frozen fractions because the heterogeneous freezing signal was so weak. These minerals were therefore also measured in 50 wt% suspensions. Onset temperatures were evaluated as well and designated by black stars in Fig. 4. We refer to the reference minerals with the identification given by the Institute of Geochemistry and Petrology even if the XRD analysis revealed a different composition.

The DSC thermograms of the emulsion freezing experiments for the minerals calcite, dolomite, dolostone, and ankerite are shown in Fig. 5a. Note that the dolostone sample was identified as dolomite by XRD. The 10% heterogeneously frozen fractions are all below 240 K. These samples also exhibit low bulk freezing temperatures between 251 and 260 K with most





freezing events around 254 K. Table 5 shows that less than 1% of the particles were active as IN. Low ice nucleation activity of calcite is in accordance with Atkinson et al. (2013). The DSC thermogram of the ankerite sample is very similar as the one of calcite, in agreement with the XRD identification of this sample as calcite. Quartz showed a weak heterogeneous signal at 244 – 247 K for 10% heterogeneously frozen fraction, at 243 – 246 K for 25% heterogeneously frozen fraction and at 240 –

245 K for 50% heterogeneously frozen fraction but with a low active particle fraction of $f_{act}$ = 0.01. These results are in qualitative agreement with Atkinson et al. (2013), who found heterogeneous freezing up to 247 K. Zolles et al. (2015) found considerably different nucleation efficiencies for three different pure alpha quartz samples with median freezing temperatures $T_{50}$ of 235, 239, and 249 K. The reason for these large differences might be due to different numbers of defects present on the surface of the particles (Zolles et al., 2015). For the muscovite (mica) sample no heterogeneous freezing was observed in

emulsion experiments. This is consistent with Atkinson et al. (2013) who found hardly any ice nucleation activity for their mica sample.

Figure 5b shows the thermograms for the emulsion experiments for all examined K-feldspars for concentrations ranging from 0.5 to 10 wt%. Interestingly, there are significant differences between their ice nucleation activities. Microcline N is the most effective for emulsions as well as for bulk freezing experiments. It showed 10% heterogeneously frozen fractions at 250 – 253

K, 25% at 249 – 252 K and 50% at 248 – 251 K. Bulk freezing occurred above 270 K, which is exceptionally high compared with all other samples. Microcline E showed 10% heterogeneously frozen fractions at 249 – 252 K, 25% at 248 – 251 K and 50% at 246 – 249 K. Bulk freezing occurred between 265 – 268 K. The microclines have also high active particle fractions of $f_{act}$ = 0.54 (microcline N) and 0.64 (microcline E). The orthoclase 2 sample proved to be an efficient IN with $f_{act}$ = 0.4, in accordance with its XRD analysis which identified it as 78% microcline with a minor share of plagioclase (13%). The 10%

heterogeneously frozen fraction occurred at 247 – 251 K, 25% at 246 – 250 K, 50% at 244.5 – 248.5 K and bulk freezing at 263 – 268 K. The sanidine sample proved to be much worse at nucleating ice than the microclines with heterogeneously frozen fractions of 10% below 241 K. Bulk freezing occurred between 252 – 256 K and $f_{act}$ was only 0.03. Our XRD analysis showed that adularia 1 consists of 100% sanidine. Nevertheless, the DSC thermograms are different from the ones of the sanidine sample. Adularia 1 was superior as an IN with $f_{act}$ = 0.23 and 10% heterogeneously frozen fraction up to 248 K.

Orthoclase 1 contains sanidine as the major component (77%) and plagioclase (15%), quartz and illite (both 4%) as minor components. The active particle fraction is higher than the one of the sanidine reference sample. DSC thermograms of the orthoclase 1 sample show two peaks. The one at lower temperatures prevails at lower suspension concentration and can be ascribed to the sanidine component. The one at higher temperatures becomes dominant at 5 – 10 wt% suspension concentration and might at least be partly due to the illite component in the sample. Adularia 2, which was indeed adularia

based on the XRD analysis, exhibited 10%, 25% and 50% heterogeneously frozen fractions below 245.7 K and bulk freezing at 252 – 258 K. The active particle fraction is 0.008.

Figure 5c shows the thermograms for the emulsion experiments for all examined (Na, Ca)-feldspars for concentrations ranging from 0.5 to 10 wt%. Plagioclase shows low freezing temperatures (below 240 K) for emulsion experiments and a





strong scattering of bulk freezing temperatures from quite low (251 K) up to quite high (263 K). The albite (pericline) and the labradorite samples show similarly low freezing efficiency as the plagioclase sample. This is in accordance with our XRD analysis, which found 100% plagioclase in the labradorite sample and 84% in the albite (pericline) sample, with the rest consisting of quartz. For the anorthite sample, in addition to plagioclase as the main component (59%), 12% quartz, 8%

muscovite and 5% kaolinite were identified as most abundant minor components. These minor components are most probably the reason for the ice nucleation activity reaching to a higher temperature compared with the other (Na,Ca)-feldspar samples. These measurements confirm the findings of Atkinson et al. (2013) that (Na,Ca)-feldspars are less efficient IN than K-feldspars. Compared with the (Na,Ca)-feldspars, all K-feldspars proved to be quite good IN. Nevertheless, the differences between them are significant with microcline being superior to the others. Therefore, it is not sufficient to discriminate just

between K-felspars and (Na, Ca)-feldspars but the specific crystal structure of the feldspars has to be considered as well.

Clay minerals have proven to be efficient IN at T < 246 K (Pinti et al., 2012). In Fig. 4 and Table 5 the values for the clay minerals measured by Pinti et al. (2012) are also given. The DSC thermograms shown in Pinti et al. (2012) were re-evaluated using the procedures outlined in Sect. 3 to obtain consistency with the new data presented in this study. The heterogeneously frozen fractions reported here are the sum of the heterogeneous signals without differentiating between standard and special

peaks. Four different samples of montmorillonite (a smectite) have been investigated by Pinti et al. (2012): M KSF and M K-10 from Sigma-Aldrich and STx-1b and SWy-2 from the Clay Mineral Society (CMS). None of these samples is mineralogical pure (Atkinson et al., 2013). M KSF and M K-10 contain illite, quartz, and feldspars as additional components, M Swy-2 contains 8% quartz and 16% feldspars as minor components. Montmorillonite can account for ice nucleation up to 240 K (freezing onset of standard sites) with peak maxima at 236 – 237 K. M SWy-2 shows a strong special peak for higher

suspension concentrations, which might arise from the feldspar component. The kaolinite samples KGa-2 and KGa-1b from CMS proved to be almost mineralogical pure kaolinite, while kaolinite from Sigma Aldrich (K-SA) consists of only 83% kaolinite with additional shares of 5% illite and 5% K-feldspar (Atkinson et al., 2013). Kga-1b and Kga-2 nucleate ice up to 242 K with $f_{act}$ = 0.041 and 0.11, respectively, and show peak maxima at 237 – 238 K (this work and Pinti et al., 2012). While the kaolinite fraction gives rise to the average freezing peak in the DSC thermograms of K-SA, the minor fractions of illite

and K-feldspar can account for the special freezing peak which appears for higher suspension concentration in the DSC thermogram. Illite references (Ill SE and Ill NX) exhibit broad peaks with maxima at 239 – 242 K and can account for heterogeneous freezing up to 246 K with an active fraction of 0.18 and 0.12, respectively. Pinti et al. (2012) and Atkinson et al. (2013) report illite as major component of illite NX and illite SE together with minor fractions of feldspar, however, without specifying the exact type of feldspar. The heterogeneous freezing signal in the DSC thermograms of illite SE and illite

NX arises most probably from the main illite component of the sample. The minor feldspar component might be responsible for the tail to high temperature of the freezing peak of illite SE.



## 6 Discussion

### 6.1 Number size distributions

The number size distributions of the natural dust samples peak between 216 nm – 479 nm (see Table 1). Except for the commercially available ATD, the only preprocessing applied to the samples was sieving with a grid of 32 µm. This led to a strong reduction of the sample volume (by a factor of 100 – 1000), especially for the dune samples from Qatar and Oman. Although the sieved samples retained particles with diameters up to 32 µm, the remaining fraction is clearly dominated by particles in the submicron range in terms of number. The dune samples from Qatar and Oman might be coarser, but no size distributions could be obtained because not enough sample material remained after sieving. Milling of the reference minerals resulted in similar size distributions to those observed for the natural samples. The finest powder was obtained for calcite (size distributions peaking at 283 nm diameter), the coarsest one for labradorite (467 nm).

Airborne mineral dust particles cover a large size range from less than 0.1 to more than 100 µm (Maring et al., 2003). In dust plumes also particles with diameters up to 30 – 40 µm are transported over long distances (Wagner et al., 2009; Reid et al., 2013; Kandler et al., 2011a; 2011b). Ground-based and airborne measurements of aerosols show that mineral dust number size distributions are dominated by submicron particles for turbid to clear conditions and particles < 2 µm in diameter in dust plumes (Kandler et al. 2009; 2011a; 2011b). Niedermeier et al. (2014) derived a lognormal number size distribution for mineral dust transported to the Cape Verde that peaks at 800 nm. This value is higher than the ones of the natural dust samples of this study, which were collected from the ground. Despite this latter discrepancy, the natural dust samples obtained in this study by sieving with a 32 µm grid reflect the number size distributions of airborne mineral dusts reasonably well in view of the overall variability given by source regions, emission and transport processes.

### 6.2 Correlation between mineralogical composition and freezing behaviour of the natural dust samples

#### 6.2.1 Emulsion experiments

In the following, we compare the freezing characteristics of the reference samples with the ones of the dust samples to investigate whether the freezing behaviour of the natural dusts can be explained in terms of their mineralogy. For this qualitative analysis, we use the active particle fractions $f_{act}$ listed in Tables 2 and 5 and the freezing characteristics observed in the DSC thermograms.

**Reference minerals.** Quartz shows ice nucleation activity up to 247 K, however with a low active particle fraction, $f_{act}$, of only about 0.01. Montmorillonites can account for freezing up to 240 K with $f_{act}$ of around 0.09, if one takes M STx-1b as most representative, kaolinite for freezing up to 242 K with $f_{act} = 0.04 – 0.11$, and illite for freezing up to 246 K with $f_{act} = 0.12 – 0.18$. The freezing behaviour of (Na, Ca)-feldspars is best represented by plagioclase, which can account for immersion freezing up to 240 K but only with $f_{act} = 0.006$. The microcline reference minerals from Elba (E) and Namibia (N) have a freezing peak with a maximum at 250 – 251 K and active particle fractions of 0.64 and 0.54, respectively. Sanidine can account for freezing up to 242 K with $f_{act} = 0.03$ and shows a peak maximum at 238 – 239 K. Adularia can account for freezing up to 246 K with $f_{act} = 0.008$ and shows a peak maximum at 239 – 240 K. Since no orthoclase was identified for the




orthoclase minerals, and hardly any ankerite in the ankerite sample, we cannot use them as references. We consider the ice nucleation activity of dolomite, dolostone, ankerite, muscovite and calcite as too low to contribute significantly to immersion freezing of the natural dusts. Finally, some of the identified minerals in the natural dusts are water soluble and lead to a freezing point depression when they dissolve in the suspensions. These minerals are halite (NaCl), thenardite ($Na_2SO_4$), and

trona ($Na_3(CO_3)(HCO_3) \times 2H_2O$).

Inspection of the natural dust samples with respect to these reference minerals reveals the following dependencies:

(1) **The Antarctica sample** has an active particle fraction $f_{act}$ = 0.067 and two distinct freezing peaks in the DSC thermograms with maxima at 241– 243 K and 251 – 252 K, which are present already at the lowest suspension concentrations. The plagioclase component (21.7%) can account for the freezing peak at lower temperature, the

microcline component (12.8%) for the one at higher temperature.

(2) **Arizona Test Dust** is abundant in mineralogical components that are active as IN and the 5 wt% suspension has indeed a high active particle fraction $f_{act}$ = 0.51. At lower suspension concentrations illite, kaolinite, sanidine, microcline, and smectite contribute to the DSC signal, at high suspension concentrations the maximum of the freezing peak shifts to 250 – 251 K and can be attributed to the microcline component.

(3) **The Bolivia sample** has an active particle fraction $f_{act}$ = 0.025. Its DSC thermogram can be explained by the presence of smectite (montmorillonite), kaolinite and plagioclase at lower freezing temperatures and for lower concentrations. No mineralogical component could be identified which would account for the peaks at higher temperature which appear at higher suspension concentrations.

(4) **The Etosha sample** has an active particle fraction $f_{act}$ = 0.32 and a broad heterogeneous freezing peak with onset at 247

K and two maxima at about 242.5 K and 239.5 K. None of the identified minerals can explain this high ice nucleation efficiency. The Etosha sample has a high share of ankerite (22.8%), but as the reference stones turned out to be calcite, we lack an ankerite reference for comparison with the dust sample. Ankerite is mineralogically similar to calcite. Since calcite is hardly active as IN, the same might be true for ankerite.

(5) **The Hoggar Mountain dust sample** has $f_{act}$ = 0.063 in immersion freezing mode, which can be explained by the

presence of sanidine, kaolinite, smectite (montmorillonite) and illite for freezing at lower temperatures and lower suspension concentrations. Illite seems to dominate heterogeneous freezing for higher temperatures and concentrations. Pinti et al. (2012) already pointed out the similarity between the freezing signal of Hoggar Mountain dust and illite with slightly higher onset freezing temperatures for Hoggar Mountain dust than for illite.

(6) **The Israel samples** show $f_{act}$ = 0.12 – 0.14, despite their high shares of calcite (65 – 68%). The heterogeneous freezing

signal seems to be a superposition of ice nucleation induced by the many minor components present in the samples. The peak at 244 K appearing at higher suspension concentration can be ascribed to illite. However, the amount of illite measured in the Israel samples seems too low to fully explain this peak.



(7) **The Makgadikgadi samples** contain water soluble components such as halite, trona, and thenardite, which dissolve in the aqueous suspensions and lead to a freezing point depression. This explains the shift of the homogeneous freezing peak to lower temperatures with increasing suspension concentration. Makgadikgadi A contains several components that can account partly for the quite high active particle fraction $f_{act}$ = 0.23 (quartz, plagioclase, smectite). With

increasing suspension concentrations additional peaks at higher temperature appear in the DSC thermograms. This seems to indicate that the DSC signal can be attributed to ice nucleation by major components at lower suspension concentrations and by minor components at higher concentration. Although most identified components of Makgadikgadi A show activity as IN, they cannot fully explain the observed peaks at higher suspension concentrations in the thermograms. The sample contains 3% adularia, which can only partly explain the signal peaking at 245 K.

Similarly, ice nucleation by the reference minerals that were identified for Makgadikgadi B cannot fully explain freezing above 242 K observed at higher suspension concentrations for this sample. Makgadikgadi C has a quite low active fraction $f_{act}$ = 0.037, in agreement with its quite high share of calcite (43%), which does not contribute to heterogeneous freezing. The presence of kaolinite (10%) and smectite (20%) can explain the DSC signal peaking at 237 K. The shoulder at 242 K observed for the highest suspension concentration can only be partly explained by the adularia

component (3%).

(8) **The dune dusts from Oman and Qatar** contain the main mineralogical components quartz, calcite, and dolomite, which show hardly any activity as IN. This is consistent with the weak heterogeneous signal observed for these samples. The higher ice nucleation activity observed for the Qatar sample can be explained by the presence of microcline.

In summary, the mineralogical composition can qualitatively explain the observed freezing behaviour of 5 of the investigated

12 natural dust samples (Antarctica, ATD, Hoggar Mountain, Oman, and Qatar dusts), and partly for six samples (Bolivia, Israel 1 and 2, Makgadikgadi A, B, and C dusts). There were no mineral components identified for the Etosha sample that would explain its high freezing efficiency. This shows that the mineralogical composition is a major determinant of the ice nucleation ability of natural mineral dust samples, but cannot explain it to the full extent. Assuming that the mineralogical composition was identified correctly, additional factors such as mixing state, morphology, and surface defects might also

influence the nucleation ability (see Sect. 6.4 for further discussion).

### 6.2.2 Bulk experiments

Natural dust samples show bulk freezing temperatures in a compact range from 255 – 265 K, while reference minerals span a broad range from 250 – 272 K (Fig. 4). Since one nucleation event initiates the freezing of the whole sample, impurities well below the detection limit of the XRD measurements might cause freezing. Nevertheless, there is a weak correlation between

bulk and emulsion freezing temperatures. Figure 6 depicts this correlation for the 10% heterogeneously frozen fraction ($T_{het,10\%}$) in the emulsion experiments performed with 0.5 wt% and 10 wt% suspensions. Samples with high freezing temperatures in emulsion experiments exhibit in general also high bulk freezing temperatures. This is the case for the natural samples (Fig. 6a) as well as for the reference minerals (Fig. 6b). An outlier is montmorillonite K-10 with high bulk freezing



but low emulsion freezing temperatures. The presence of a correlation suggests that the best sites which are responsible for freezing of bulk samples are not just impurities but related to the mineralogical composition. Samples containing microcline (Antarctica, ATD, microclines E and N, and orthoclase 2) are among the ones with the highest bulk freezing and emulsion freezing temperatures. Calcite, dolomite, and dolostone with low ice nucleation activity in emulsion experiments also show

low bulk freezing temperatures.

### 6.3 Active particle fraction $f_{act}$

The active particle fractions range from $f_{act} = 0.025 – 0.32$ (Table 2) for the natural dust samples excluding ATD and from $f_{act} = 0.0004 – 0.64$ for the reference minerals (Table 5). The minerals calcite, dolomite and dolostone are virtually inactive as IN and the heterogeneous freezing observed for these samples could also be due to impurities instead of the minerals themselves.

Microclines proved to be exceptionally good as IN with $f_{act} = 0.54$ for microcline E and $f_{act} = 0.64$ for microcline N. Considering the large uncertainties associated with $f_{act}$ (see Appendix A) even all particles might be active as IN. This is supported by the DSC thermograms featuring the complete decline of the heterogeneous signal before homogeneous freezing sets in. Most other reference minerals have $f_{act} = 0.01 – 0.1$ indicating the presence of particles that do not act as IN. Accounting for a potential low bias because large particles may be missed when size distributions are determined by

SMPS/APS (see Appendix A), $f_{act}$ could rise to $0.05 – 0.5$ but should remain clearly below unity. A value of $f_{act}$ significantly below one is further supported by the observation that heterogeneous nucleation is still ongoing when homogeneous freezing sets in.

Assuming ice nucleation to occur on active sites, whose occurrence can be described by a probability density as a function of surface area (e.g. Marcolli et al., 2007; Lüönd et al., 2010), the inactive particles are more likely the smallest ones. Our data

cannot confirm or reject this relation because in our experiments the whole distribution of particles is investigated without any extra information on size dependence. There is also hardly any direct proof for such a relationship from other studies on immersion freezing. The mineral dusts chosen to study size-selected particles of different diameters were mostly mixtures of minerals, so that different particle sizes could correlate with different mineralogical composition.

Such a case is ATD as it is a complex mixture of minerals (see Table 3). With the Leipzig Aerosol Cloud Interaction

Simulator (LACIS), Niedermeier et al. (2010, 2011) investigated ice nucleation in immersion mode for 300 nm ATD particles and determined ice active fractions of $f_{act} = 0.04$ at 239 K. This fraction increased to 0.3 at 235 K and reached 1.0 only at 234 K due to homogeneous ice nucleation. In a continuous flow diffusion chamber (CFDC), Hoyle et al. (2011) measured maximum activated fractions of $f_{act} = 0.8$ for 800 nm diameter ATD particles before homogeneous ice nucleation set in. This result was confirmed by Nagare et al. (2016), who measured $f_{act} \leq 1$ before homogeneous freezing set in for 800 nm ATD

particles with the same CFDC. The active site parameterization developed by Marcolli et al. (2007) based on DSC experiments agrees well with these experiments. It further predicts that ATD particles need to be larger than 200 nm to bear an active site inducing heterogeneous freezing before homogeneous ice nucleation sets in. A 300 nm diameter ATD particle should contain on average an active site that induces heterogeneous freezing within 10 s at 237 K. In this study, the ATD





sample was used without any size selection, yielding $f_{act}$ = 0.51. A value of $f_{act}$ significantly below 1 is confirmed by the fact that the heterogeneous signal is still present when homogeneous ice nucleation sets in. The size distribution determined for ATD peaks at 354 nm. The fraction of particles with diameters below 300 nm is 0.393, the fraction below 200 nm is 0.174. This explains $f_{act}$ < 1 in our experiments.

Kaolinite KGa-1b from CMS has been investigated by several groups. Re-evaluating the data presented in Pinti et al. (2012), yields $f_{act}$ = 0.041. If we account for a low bias, this number might rise at maximum to $f_{act}$ = 0.2. A value of $f_{act}$ well below 1 is in accordance with the DSC thermogram (Pinti et al., 2012) with an overlap of the homogeneous and heterogeneous ice freezing peaks, showing that heterogeneous nucleation is still ongoing when homogeneous ice nucleation sets in. Wex et al. (2014) investigated 300 nm KGa-1b particles and determined a nucleation rate coefficient. If their parameterization is applied

to 300 nm particles and a residence time of 10 s, $f_{act}$ = 1 is expected at 236 K when homogeneous nucleation sets in. Active fractions of 0.5 and 0.1 are expected for 200 nm and 100 nm particles, respectively. The size distribution of KGa-1b peaks at 302 nm. The share of particles < 200 nm and < 100 nm is 0.241 and 0.0294, respectively. Therefore, the active particle fraction of KGa-1b observed in our experiments seems rather low when compared with the parameterization of Wex et al. (2014).

In experiments with a CFDC, Lüönd et al. (2010) measured active particle fractions of almost 1 for droplets containing one kaolinite (K-SA) particle with diameter of 200 nm and 400 nm, and 100% when droplets contained an 800 nm particle. These numbers are higher than $f_{act}$ = 0.13 that we determined for K-SA in our experiments. Wex et al. (2014) observed frozen fractions up to 0.5 for 700 nm K-SA particles. Hartmann et al. (2016) determined frozen fractions at 236 K of 0.1 for 300 nm and of 0.3 for 700 nm K-SA particles and they found an exponential increase of the freezing probability with the increase of

surface area of K-SA present in the droplet by comparing 300, 700, and 1000 nm particles. These results are in accordance with our active IN fraction of 0.13 determined for K-SA when taking measurement uncertainties and the influence of the time available for nucleation into account (Welti et al., 2012).

In summary, it is very likely that mineral particles are present in samples of mineral dusts that are inactive as IN. For mineralogically pure samples, the inactive particle fraction seems to correlate with particle size. For natural dusts, which are

mixtures of minerals, this fraction probably correlates with the mineralogical composition because the analysis of the reference samples has shown a great variability of ice nucleation activity for different minerals. However, there are also discrepancies in the active particle fractions, when these are determined with different instruments. These discrepancies must be due to systematic errors, which are currently not well-understood and are not taken into account in any error estimates.

### 6.4 Importance of mineralogy vs morphology and surface structure for ice nucleation ability

In this study, we compare freezing characteristics of natural dust samples with those of reference samples. The reference samples were powders obtained from milling of mineral stones. To investigate the influence of milling, we compare the < 32 µm fraction of the Antarctica sample, once in its original state and once milled (see Sect. 4.1, Table 2, and Fig. 3a). No



significant change in terms of active particle fraction $f_{act}$ can be observed, when comparing the results for the sieved Antarctica sample with the results for the milled Antarctica sample. The 10, 25 and 50% heterogeneously frozen volume is slightly higher for the milled Antarctica sample. An explanation for this could be that the microcline concentration in terms of number of particles might increase due to milling, since feldspar particles (amongst them microcline) are typically larger than,

e.g., clay mineral particles. The milling of the sample reduced the mode diameter $d_m$ of the lognormal distribution from $383 \pm 3$ nm to $288 \pm 0.8$ nm. Comparison of the thermograms shown in Fig. 3a shows that milling indeed influences the freezing behaviour of the sample. Milling of the sample leads to a shift of the main peak to higher temperature and the disappearance of the high temperature peak at about 253 K. A part of these changes might be attributed to the increased occupation of droplets by particles since the number of particles per mass increased (see Table 2). However, the disappearance of the high

temperature peak points to additional surface modifications. This shows that physical treatment like milling influences the freezing behaviour of samples in a complex way. Hiranuma et al. (2015) found a higher freezing efficiency for milled hematite particles. Similarly, Zolles et al. (2015) found an increase of freezing temperature by milling for quartz samples and a marginal increase for feldspar samples.

For some natural dust samples (Bolivia, Makgadikgadi A and B) freezing peaks observed at higher temperatures for higher

suspension concentrations cannot be explained by mineralogical composition. This raises the question about the importance of morphology and the surface structure. These two aspects cannot be investigated by XRD measurements. While the milled reference samples should be primary particles with only a small fraction of aggregated particles, natural samples might show higher contributions of aggregates composed of different minerals. Single particle analysis by scanning electron microscopy (SEM) and X-ray fluorescence (XRF) of airborne and surface collected dust particles during PRIDE showed that 50% of all

particles were present in some form of aggregates of different minerals (Reid et al., 2003). The larger clay minerals were usually found to be carrying smaller particles of e.g. iron oxide (Reid et al., 2013). Kandler et al. (2011a) found that aggregates were least frequent for particles in the submicron size range, since they are in the same size range as the primary grains. The aggregate structure became more complex for particles between 1 and 2.5 μm and most particles between 2.5 and 10 μm had aggregated structures. If the contact lines between aggregates performed as preferential sites for ice nucleation, this

might explain the higher nucleation ability of some natural dust samples compared with the reference samples they are composed of. In addition, active sites such as steps, cracks and cavities (Kulkarni and Dobbie, 2010; Hiranuma et al., 2014; Zolles et al., 2015) might be more frequent in naturally aged samples collected in deserts than in milled stones. Moreover, the presence of biological material has been discussed to increase the ice nucleation ability of dusts (DeMott et al., 2003; Baker et al., 2005; Pratt et al., 2009; Conen et al., 2011; Hallar et al., 2011; Creamean et al., 2013). However, the content of biological

material should be small for samples collected in deserts.



## 7 Atmospheric implications and implementation into models

We consider the natural dust samples investigated here to be representative of atmospheric mineral dust. Most of them were collected from the ground in dust source regions that contribute frequently to the airborne long range transport. The sieving of the samples with a 32 µm grid rendered them comparable in size with atmospheric mineral dusts, but left the particles otherwise unchanged.

A surprising finding is that the distribution of freezing temperatures of natural dusts is much more compact than that of the reference minerals for both emulsion measurements (characterizing the "typical IN") and bulk measurements (characterizing the "best IN"). The freezing temperatures $T_{\text{het},10\%}$ of natural dusts with 5 wt% suspension concentration span about 6 K, but those of reference minerals about 12 K (see Fig. 4). Therefore, the reference samples extend far beyond the range covered by the natural dusts in terms of high as well as low IN efficiencies. The dust mixing and/or dust aging (e.g. by coatings) in the natural environment appears to be so strong that extreme IN efficiencies (high or low) do not develop. This should help formulating parameterizations for ice nucleation under atmospheric conditions (which goes beyond the scope of this paper).

However, mineralogical composition does matter. Quartz particles were present in all samples (cp. Table 3) of the nine investigated regions (locations 1 – 9 in Fig. 2). Further, calcite, (Na, Ca)-feldspars (plagioclase) and clay minerals (mainly smectite) were present in seven of the nine regions. K-feldspars (adularia, microcline, orthoclase and sanidine), were identified still in six regions, however, only the Antarctic, ATD and Qatar dune samples contained microcline, and the latter only as a minor fraction. These compositions compare very well with the ones of atmospheric mineral dusts compiled by Murray et al. (2012), confirming their assertion that mineral dusts cannot fully account for the high freezing temperatures (above 261 K) observed in the atmosphere. A notable exception might be microcline-containing samples, for which the mineral reference sample microcline N (see Fig. 4) shows bulk freezing even above 270 K. However, out of the 10 investigated samples from dust source regions, microcline was only detected in the Qatar dune sample, and then only as a minor component (4%). Curiously, this sample did not exhibit a particularly high ice nucleation efficiency (Fig. 3). If microcline turned out to be more common in aerosols than suggested by the samples collected from dust source regions in this study, microcline particles could be relevant for cloud glaciation at temperatures above 260 K. Though from the present study this seems to be not a very likely scenario. In order to investigate this better, atmospheric aerosol samples would need to be analysed with respect to microcline using XRD.

Evaluation of the active particle fraction $f_{\text{act}}$ showed that only a part of the mineral dust particles is active as IN. The inactive fraction most probably consists of particles, which are composed of inactive minerals such as calcite or muscovite. These particles might at the same time belong to the fraction of particles with smaller size, because larger particles are usually aggregates of different minerals (Reid et al., 2003; Kandler et al., 2011a) and thus more likely contain a mineral that is an active IN. A size dependency is also in accordance with DeMott et al. (2010), who found that the concentrations of IN active at mixed-phase cloud conditions can be related to the number concentrations of particles larger than 0.5 µm in diameter. Size-selective ice nucleation experiments with pure mineral samples, such as kaolinite, show that larger particles are indeed more





effective IN than smaller ones. This justifies a parameterization of immersion freezing based on particle surface area, as developed by Niemand et al. (2012). The good correlation between mineralogical composition and freezing behaviour suggests that more sophisticated parameterizations should rely on the mineralogical composition based on a source scheme of dust emissions as done by Hoose et al. (2008).

## 8 Summary and conclusions

Natural dusts and milled reference minerals were analysed with the objective to investigate whether their ice nucleation activity shows significant differences between different source regions and whether the freezing behaviour can be related to the mineralogical composition. The natural dust samples consisted of calcite, quartz, clay minerals, K-feldspars and (Na, Ca)-
feldspars as major mineralogical components, which is in good agreement with the mineralogical composition of atmospheric mineral dusts. With number distributions that peak for diameters < 1 µm, they are also comparable in size.

The ice nucleation ability of the reference minerals show large variations, much larger than found for the natural dusts investigated in this work. Calcite, dolomite, dolostone and muscovite seem to induce hardly any freezing. For these minerals, ice nucleation, if present, might not be related to the mineral composition but controlled by impurities. Microcline is able to
induce freezing at higher temperatures than all other investigated minerals. In addition, more or less all particles in the two investigated microcline samples are active as IN. This makes this mineral an exceptionally good IN and superior to all other analysed K-feldspars, (Na, Ca)-feldspars and the clay minerals.

The XRD analysis of the natural dust samples showed that quartz particles were present in dusts from all source regions. (Na, Ca)-feldspars (plagioclase) and clay minerals (mainly smectite) were present in dusts from most source regions. K-feldspars
were identified in several source regions, however, only the Qatar dune sample contained microcline as a minor fraction – besides ATD and Antarctica samples, which are not considered typical source regions.

The natural dust samples show very similar freezing temperatures except Antarctica and ATD. These two samples were not collected in source regions of mineral dust aerosols but included as examples of very remote regions or of commonly used reference dust. For all natural dust samples without Antarctica and ATD, 10% heterogeneously frozen fraction is realized
between 244 and 250 K. The 25% heterogeneously frozen fraction is between 242 and 246 K, and the 50% heterogeneously frozen fraction is between 239 and 244 K. Bulk freezing occurred between 255 and 265 K. The natural dust samples show active particle fractions $f_{act} = 0.025 – 0.32$. Taking a potential low bias in our evaluation into account, these numbers might rise to $f_{act} = 0.1 – 1$. Active particle fractions significantly below 1 are expected considering the share of very small particles and inactive mineral components present in the dusts.

Qualitatively, the mineralogical composition can fully explain the observed freezing behaviour of 5 of the investigated 12 natural dust samples, and partly for 6 samples. Only for the Etosha sample no mineral components could be identified that would explain its high freezing efficiency. This shows that in general the mineralogical composition is a major determinant of





the ice nucleation ability of natural mineral dust samples, but cannot entirely explain it. Agglomeration of particles and surface erosion could lead to additional changes in active sites of natural dusts, either enhancing or reducing their efficiency, in comparison to freshly milled reference samples. Comparison of the DSC thermograms of only sieved and additionally milled Antarctic dust shows that milling indeed influences the freezing behaviour and possibly the surface structure of the sample.

The findings of this study are in agreement with previous work that the mineral dusts can hardly account for the glaciation of clouds observed at the highest temperatures in the atmosphere. A single notable exception is microcline, for which the temperature of 50% heterogeneously frozen fraction occurs above 245 K for all concentrations. For the microcline from Namibia, bulk freezing temperatures were even above 270 K. If microcline turns out to be more common in atmospheric mineral dusts than suggested by the mineralogical composition of the natural dust samples investigated here, this conclusion could be revised. To resolve this question, analysis of dust samples with XRD is necessary.

## Appendix A

## Uncertainties of the fraction of active IN

Different kinds of uncertainties concerning the calculation of the active particle fraction $f_{act}$ are presented in this section.

### A1 Uncertainty in calculation of size distribution

The uncertainty stemming from the size distribution of the dust particles is the largest one. Due to the fact, that particles > 1 μm tend to sediment quite fast, larger particles might not be measured appropriately with the APS. Even if the number of particles with diameter >1 μm is small compared with the total number of particles, the contribution to the mass can be significant and therefore the estimated number of particles for a given mass can change drastically. To estimate this error, two different measurement techniques were used to determine the size distribution of the Hoggar Mountain dust sample, namely SMPS/APS and the evaluation of electron microscope (EM) images (Pinti et al., 2012). The SMPS/APS measurements gave a factor 4.4 higher particle numbers per mass than the EM method. If we assume that all the particles > 16 μm diameter sediment too fast to be pipetted into the emulsion, this factor reduces to 2.8. To be consistent with the evaluation of the other samples, the SMPS/APS measurements were used to calculate $f_{act}$ in this study. However, we think that the EM evaluation represents better the coarse fraction present in the sample, which would lead to a low bias of $f_{act}$ listed in Tables 2 and 5.

### A2 Uncertainty in the calculation of water droplet size distribution

The second largest uncertainty concerns the stability of the size distribution of the water droplets in the emulsion, which is strongly influenced by the homogeneity of the mineral oil / lanolin mixture. The homogeneity decreases with time and can be restored by heating up and mixing the mineral oil / lanolin mixture again. To estimate this uncertainty, samples with the same dust and the same concentration measured at different times (some month in between) were compared. Over many months $f_{act}$ varies at most by a factor of two due to the changes in the water droplet size distribution. This agrees with the comparison of the droplet size distributions measured at different times.



Figure 1 shows that no water droplets larger than 10 μm were found in the optical images of the emulsions. However, extrapolating the lognormal distribution fitted to the volume size distribution, larger droplets should also be present. This introduces another systematic uncertainty to our evaluation. To quantify this uncertainty we extrapolated the volume size distribution of the droplets using the lognormal distribution. Taking the thus obtained fraction of particles >10 μm into account would reduce the active particle fraction by less than 10%. We also considered the error because droplets with diameters < 0.3 µm are below the detection limit of the microscope. We estimated that the contribution of these small water droplets to the whole water volume in the emulsion should be 3% or less. For measurements with less than 80% heterogeneously frozen volume this changes the number of active particles $f_{act}$ by a factor of < 1.2. For heterogeneously frozen fractions of 90% and 95%, this would increase $f_{act}$ by a factor of 1.4 and 2, respectively. Therefore this uncertainty seems to influence mainly the samples with high heterogeneously frozen fractions, namely the two microcline samples, the milled Antarctica sample and the ATD sample.

**A3 Uncertainty in the separation of heterogeneous and homogeneous freezing peaks**

The separation between the heterogeneous and homogeneous freezing peaks is for DSC thermograms with a large overlap to some extent arbitrary. Therefore the heterogeneous and homogeneous freezing peaks were separated in different ways to evaluate the heterogeneously frozen fraction. For samples where the heterogeneous signal is still high when homogeneous freezing starts, this gave an uncertainty of $f_{act}$ by a factor of 1.3 for about 10% heterogeneously frozen fractions), 1.15 for heterogeneously frozen fractions between 20% and 80%, 1.3 for  90% heterogeneously frozen fractions and 1.6 for 95% heterogeneously frozen fractions. Because all samples with heterogeneously frozen fractions > 85% have clearly separated heterogeneous and homogeneous freezing peaks, the uncertainty concerning the separation of heterogeneous and homogeneous signal is smaller and therefore $f_{act}$ has an uncertainty of a factor 1.1 – 1.25 depending on the heterogeneously frozen fraction.

**A4 Total uncertainty of the active IN fraction**

All these uncertainties give an upper limit for $f_{act}$ 5.8 times (6.7 times for microcline) larger than the calculated value. Assuming that all dust particles >15 μm sediment too fast to be pipetted into the emulsions, this would reduce the correction factor to 4.2 (5.1 for microcline). The lower limit for $f_{act}$ is 2.3 times smaller than the calculated value. The correction factor for the lower limit is smaller than for the upper limit, because for the calculation of the active particle fraction the particle size distribution measured with SMPS and APS was used. This method gives more particles for a given mass than measuring and counting the particles from electron micrographs. Therefore the particle size distribution used for calculations can be assumed to give an upper limit of the number of particles present per mass.

This error estimation shows that uncertainties of $f_{act}$ increase for higher suspension concentrations. We therefore assume that the most reliable active particle fractions are determined for lower suspension concentrations. Since the heterogeneous signal for the lowest concentrations is for many samples too weak to be evaluated reliably, we take the 2 wt% suspension data as representative for $f_{act}$, since for this concentration data for almost all samples are available.



**Appendix B Uncertainties in XRD analysis of mineralogical composition**

Table B1 compares the XRD evaluations of ATD from Atkinson et al. (2013) with the one performed in this study. The differences illustrate the accuracy that can be expected for the determination of mineralogical composition by such analyses. Both identifications found the same minerals present in ATD but with differences in the relative contributions of up to 15%. Atkinson et al. (2013) found small amounts of hematite and dolomite, which were not found in this study. Small amounts of ankerite and tremolite were identified in this study but not by Atkinson et al. (2013). Ankerite and dolomite have very similar XRD diffractograms, therefore a distinction between these two minerals is (at least for small amounts) quite difficult.

**Acknowledgements**

This work was supported by the Swiss National Foundation, projects No. 200021_138039 and 200021_140663. We thank Paolo D'Odorico, Alain Jacot, Esther Mbiti, Christian Rixen, Sonja Wipf, Jens Köhler, Georg Kaser and Yvonne Boose for providing the natural dust samples. Michael Plötze, Anette Rötlisberger and Marion Rothaupt for the possibility to do XRD measurements. André Welti, Baban Nagare and Monika Kohn for providing the SMPS and the APS and the strong support during size distribution measurements. Peter Brack for providing the reference minerals. Alejandro Beltran and Wilfried Winkler for providing the infrastructure for sieving. Kurt Barmettler for providing the infrastructure to mill the stones. Moreover, we thank Ulrike Lohmann for helpful discussion.

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



**Table 1.** Lognormal size distribution parameters $d_m$ (nm) and $w$ for the analysed samples according to Eq. (1). Error ranges reflect fit uncertainties. For the Oman and Qatar samples not enough material was available to measure the size distribution.

| Category | Mineral/dust | $d_m$ (nm) | $w$ |
|---|---|---|---|
| Antarctica and ATD | Antarctica | 383 ± 3 | 0.542 ± 0.005 |
| | Antarctica milled | 288 ± 0.8 | 0.443 ± 0.002 |
| | ATD | 354 | 0.609 |
| Natural dust source regions | Bolivia | 216 ± 1.9 | 0.603 ± 0.006 |
| | Etosha | 414 ± 6 | 0.797 ± 0.008 |
| | Hoggar | 312 ± 2 | 0.609 ± 0.004 |
| | Israel 1 | 401 ± 2 | 0.554 ± 0.004 |
| | Israel 2 | 479 ± 4 | 0.545 ± 0.005 |
| | Makgadikgadi A | 354 ±3 | 0.632 ± 0.004 |
| | Makgadikgadi B | 283 ± 1.9 | 0.694 ± 0.004 |
| | Makgadikgadi C | 274 ± 2 | 0.678 ± 0.005 |
| Reference minerals | | | |
| *Naturally abundant* | Ankerite | 329 ± 1.5 | 0.592 ± 0.003 |
| | Calcite | 283 ± 3 | 0.504 ± 0.007 |
| | Dolomite | 304 ± 2 | 0.536 ± 0.005 |
| | Dolostone | 380 ± 1.7 | 0.529 ± 0.003 |
| | Muscovite | 335 ±6 | 0.558 ± 0.011 |
| | Quartz | 364.5 ± 1.8 | 0.528 ± 0.003 |
| *K-feldspars* | Adularia 1 | 349 ± 1.8 | 0.528 ± 0.004 |
| | Adularia 2 | 249 ± 0.5 | 0.544 ± 0.002 |
| | Microcline Elba | 399 ± 2 | 0.530 ± 0.003 |
| | Microcline | 412 ± 3 | 0.531 ± 0.004 |
| | Orthoclase 1 | 417 ± 3 | 0.528 ± 0.004 |
| | Orthoclase 2 | 262.8 ± 0.9 | 0.590 ± 0.002 |
| | Sanidine | 372.5 ± 1.5 | 0.524 ± 0.003 |
| *(Na-Ca)-feldspars* | Albite (Pericline) | 439 ± 3 | 0.548 ± 0.005 |
| | Anorthite | 336 ± 3 | 0.523 ± 0.005 |
| | Labradorite | 467 ± 4 | 0.528 ± 0.005 |
| | Plagioclase | 404 ± 2 | 0.499 ± 0.004 |
| *Clay minerals* | Illite NX | 354 ± 4 | 0.625 ± 0.008 |
| | Illite SE | 317 ± 3 | 0.576 ± 0.006 |
| | KGa-1b | 302 ± 4 | 0.585 ± 0.009 |
| | KGa-2 | 353 ± 4 | 0.562 ± 0.007 |
| | K-SA | 416 ± 5 | 0.609 ± 0.007 |
| | MK-10 | 395 ± 4 | 0.584 ± 0.007 |
| | MKSF | 453 ± 12 | 0.628 ± 0.016 |
| | MSTx-1b | 278 ± 4 | 0.655 ± 0.010 |
| | MSWy-2 | 279 ± 4 | 0.647 ± 0.008 |



**Table 2**. Evaluation of active particle fractions ($f_{act}$) in natural dust samples evaluated for 2 wt% suspensions, with the exception of ATD (5 wt%). $D_{p1}$: droplet diameter with on average 1 particle inside (µm); $p_{het}$: calculated heterogeneously frozen water volume fraction assuming that droplets containing particles froze heterogeneously, $p_{het,lab}$ measured heterogeneously frozen water volume fraction.

| Sample | $D_{p1}$ | $p_{het}$ | $p_{het,lab}$ | $f_{act}$ |
|---|---|---|---|---|
| Antarctica | 3.25 | 0.89 | 0.37 | 0.067 |
| Antarctica milled | 2.00 | 0.97 | 0.73 | 0.088 |
| ATD | 2.25 | 0.95 | 0.90 | 0.510 |
| Bolivia | 2.00 | 0.98 | 0.49 | 0.025 |
| Etosha | 5.50 | 0.59 | 0.33 | 0.320 |
| Hoggar | 2.75 | 0.91 | 0.43 | 0.063 |
| Israel 1 | 3.25 | 0.87 | 0.46 | 0.120 |
| Israel 2 | 3.75 | 0.80 | 0.37 | 0.140 |
| Makgadikgadi A | 3.25 | 0.86 | 0.62 | 0.230 |
| Makgadikgadi B | 3.00 | 0.89 | 0.53 | 0.120 |
| Makgadikgadi C | 2.75 | 0.91 | 0.31 | 0.037 |



**Table 3.** Mineralogical composition of natural dust samples in wt% derived from the Rietveld analysis of the X-ray diffraction (XRD) patterns. The abbreviations stand for cal: calcite; dol: dolomite; mus: muscovite; qu: quartz; adu: adularia; mic: microcline; ort: orthoclase; san: sanidine; pla: plagioclase; ill: illite; ka: kaolinite; sm: smectite (montmorillonite).

| Sample | cal | dol | mus | qu | adu | mic | ort | san | pla | ill | ka | sm | others |
|---|---|---|---|---|---|---|---|---|---|---|---|---|---|
| Antarctica | | | 18 | 24 | | 15 | 4 | | 29 | | | | biotite 2; chlorite 3; epidote 4 |
| ATD | 1 | | | 23 | 29 | 1 | 6 | | 12 | 3 | <1 | 25 | ankerite <1; tremolite <1 |
| Bolivia | 17 | | | 4 | | | | | 22 | | 4 | 50 | wollastonite 3 |
| Etosha | 29 | 27 | 10 | 1 | | | | | | | | 1 | analcime <1; ankerite 23; chabazite <1: halite <1; kyanite 2; sepiolite 4; tridymite <1 |
| Hoggar | | | | 13 | | | | 10 | 13 | 4 | 6 | 48 | biotite 4; brookite <1; fluorapatite <1; Hematite <1 |
| Israel 1 | 68 | | 6 | 8 | | | | 2 | 2 | <1 | | 6 | ankerite 8 |
| Israel 2 | 65 | | 2 | 10 | | | | 2 | 3 | 3 | | 7 | ankerite 5; fluorapatite 2; magnesite <1; |
| Makgadikgadi A | 25 | | 21 | 4 | 3 | | | | 5 | | | 23 | anatase <1; halite 2; szomolnokite 1; trona 14; virgilite <1 |
| Makgadikgadi B | 3 | | 1 | 1 | | | | <1 | | | 76 | 6 | cristobalite <1; halite 2; thenardite 10; |
| Makgadikgadi C | 43 | | | 3 | 3 | | | | | | 10 | 20 | barite <1; feruvite 1; halite 3; lepidolite 3; nontronite 5; sulfur alpha 4; thenardite 6 |
| Oman | 29 | | | 26 | | | | 5 | 12 | | | | chlorite 7 |
| Qatar | 31 | 21 | | 12 | 4 | | | | 8 | | | 13 | chlorite 2; diopside 1; ferrite magnesion <1; hematite 2; hornblende 3 |



**Table 4.** Mineralogical composition of reference samples in wt%. The abbreviations stand for cal: calcite; dol: dolomite; mus: muscovite; qu: quartz; adu: adularia; mic: microcline; ort: orthoclase; san: sanidine; pla: plagioclase; ill: illite; ka: kaolinite; sm: smectite (montmorillonite).

| Sample | cal | dol | mus | qu | adu | mic | ort | san | pla | ill | ka | sm | others |
|---|---|---|---|---|---|---|---|---|---|---|---|---|---|
| Ankerite | 99 | | | | | | | | | | | | ankerite <1; |
| Calcite | 100 | | | | | | | | | | | | |
| Dolomite | | 100 | | | | | | | | | | | |
| Dolostone | | 100 | | | | | | | | | | | |
| Muscovite | | | 100 | | | | | | | | | | |
| Quartz | | | | 100 | | | | | | | | | tungsten carbide <1 |
| Adularia 1 | | | | | | | | 100 | | | | | |
| Adularia 2 | | | | | 100 | | | | | | | | |
| Microcline E | | | | | | 90 | | | 10 | | | | |
| Microcline N | | | | <1 | | 77 | | | 22 | | | | langanite <1 |
| Orthoclase 1 | | | | 4 | | | | 77 | 15 | 4 | | | |
| Orthoclase 2 | | | | 2 | | 78 | | | 13 | | | | epidote 6; chlorite 2 |
| Sanidine | | | | | | | | 100 | | | | | |
| Albite | | | | 16 | | | | | 84 | | | | |
| Anorthite | | 1 | 8 | 12 | | | | | 59 | | 5 | | azurite <1; feruvite 2; prehnite |
| Labradorite | | | | | | | | | 100 | | | | |
| Plagioclase | | | | | | | | | 100 | | | | |



**Table 5**. Evaluation of active particle fractions ($f_{\text{act}}$) of natural dust samples applied to 2 wt% suspensions, with the exceptions of ankerite (50 wt%), dolomite (50wt%), and muscovite (10 wt%). $D_{p1}$: droplet diameter with on average 1 particle inside (µm); $p_{\text{het}}$: calculated heterogeneously frozen droplet volume fraction assuming that droplets containing particles froze heterogeneously, $p_{\text{het,lab}}$ measured heterogeneously frozen water volume fraction.

| Sample | $D_{p1}$ | $p_{\text{het}}$ | $p_{\text{het,lab}}$ | $f_{\text{act}}$ |
|---|---|---|---|---|
| Ankerite | 1 | 1.00 | 0.15 | 0.0007 |
| Calcite | 2.25 | 0.96 | - | - |
| Dolomite | 0.75 | 1.00 | 0.13 | 0.0004 |
| Dolostone | 2.75 | 0.90 | - | - |
| Muscovite | 1.5 | 0.99 | - | - |
| Quartz | 2.75 | 0.91 | 0.12 | 0.012 |
| Adularia 1 | 2.75 | 0.92 | 0.72 | 0.23 |
| Adularia 2 | 2 | 0.97 | 0.20 | 0.008 |
| Microcline E | 3.25 | 0.88 | 0.82 | 0.64 |
| Microcline N | 3.25 | 0.87 | 0.78 | 0.54 |
| Orthoclase 1 | 3.25 | 0.87 | 0.54 | 0.17 |
| Orthoclase 2 | 2.25 | 0.95 | 0.88 | 0.40 |
| Sanidine | 2.75 | 0.90 | 0.23 | 0.030 |
| Albite | 3.5 | 0.84 | 0.11 | 0.022 |
| Anorthite | 2.75 | 0.93 | 0.27 | 0.025 |
| Labradorite | 3.75 | 0.82 | 0.045 | 0.009 |
| Plagioclase | 3 | 0.89 | 0.049 | 0.0061 |
| Illite NX | 3.25 | 0.87 | 0.48 | 0.12 |
| Illite SE | 2.75 | 0.93 | 0.69 | 0.18 |
| KGa-1b | 2.5 | 0.93 | 0.37 | 0.041 |
| Kga-2 | 2.75 | 0.91 | 0.53 | 0.11 |
| K-SA | 3.75 | 0.82 | 0.4 | 0.13 |
| M K-10 | 3.25 | 0.86 | 0.32 | 0.067 |
| M KSF | 4.25 | 0.76 | 0.066 | 0.022 |
| M STx-1b | 2.75 | 0.92 | 0.52 | 0.089 |
| M Swy-2 | 2.75 | 0.93 | 0.82 | 0.40 |




**Table B1:** Comparison of mineralogy of ATD determined by XRD from Atkinson et al. (2013) and this study. The bbreviations stand for ill/mus: illite/muscovite; ka: kaolinite; qu: quartz; Kf: K-feldspars, NCf: (Na, Ca)-feldspars; cal: calcite; sm: smectite; dol: dolomite.

| Natural dusts | ill/mus | ka | qu | Kf | NCf | cal | sm | dol | others |
|---|---|---|---|---|---|---|---|---|---|
| this study | 3 | <1 | 23 | 36 | 12 | 1 | 25 | | Ankerite <1; Tremolite <1 |
| Atkinson et al. (2013) | 7.5 | 2 | 17.1 | 20.3 | 12.4 | 4.3 | 10.1 | 1.3 | Hematite 0.7 |

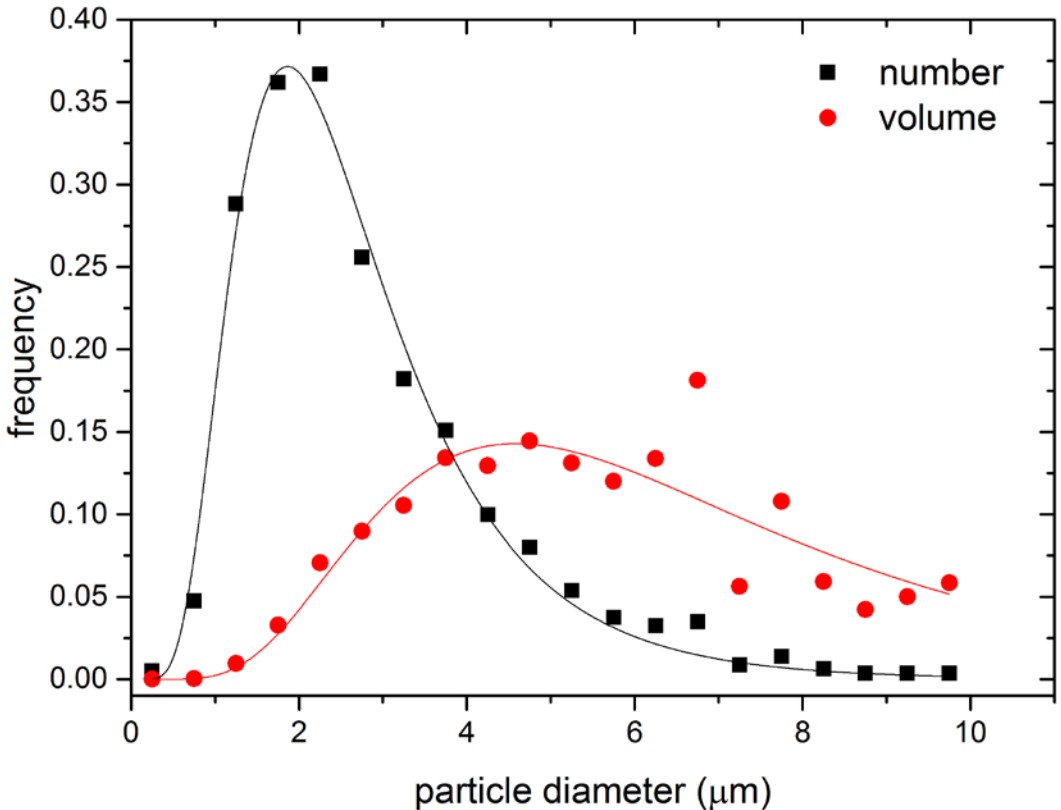

**Figure 1.** Normalized lognormal size distribution of emulsion droplets. Data points represent bin widths of 0.5 µm. The parameters for the lognormal number distribution are $d_m$ = 2.41 ± 0.04 µm and $w$ = 0.507 ± 0.014 and for the volume distribution $d_m$ = 6.1 ± 0.4 µm and $w$ = 0.53 ± 0.05.





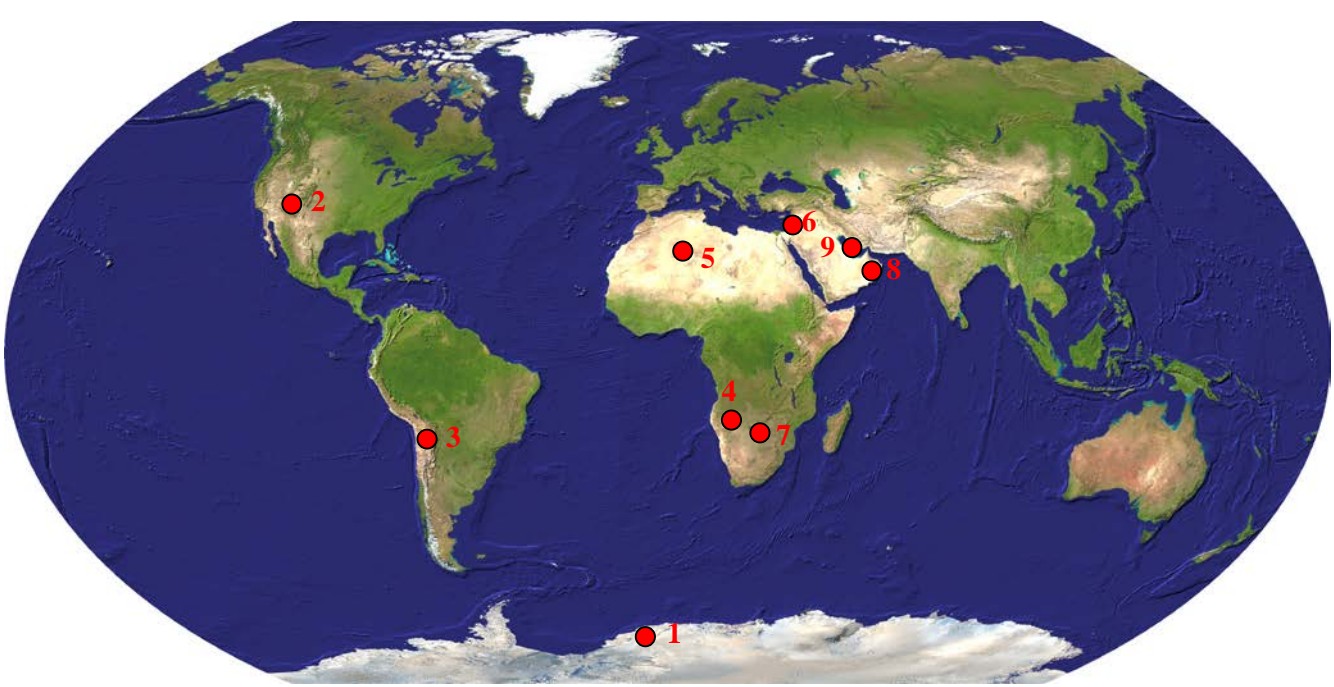

5 **Figure 2.** Collection locations of natural dust samples (in alphabetic order): (1) Antarctica, (2) Arizona, (3) Bolivia, (4) Etosha, (5) Hoggar, (6) Israel, (7) Makgadikgadi, (8) Oman, (9) Qatar.

**Figure 3a.** DSC thermograms of natural dust samples: Antarctica, Antarctica (milled), Bolivia, Etosha, Hoggar and Israel 1.





**Figure 3b.** DSC thermograms of natural dust samples: Israel 2, Makgadikgadi A, B and C, Oman and Qatar. Suspension concentrations are given in the legend.



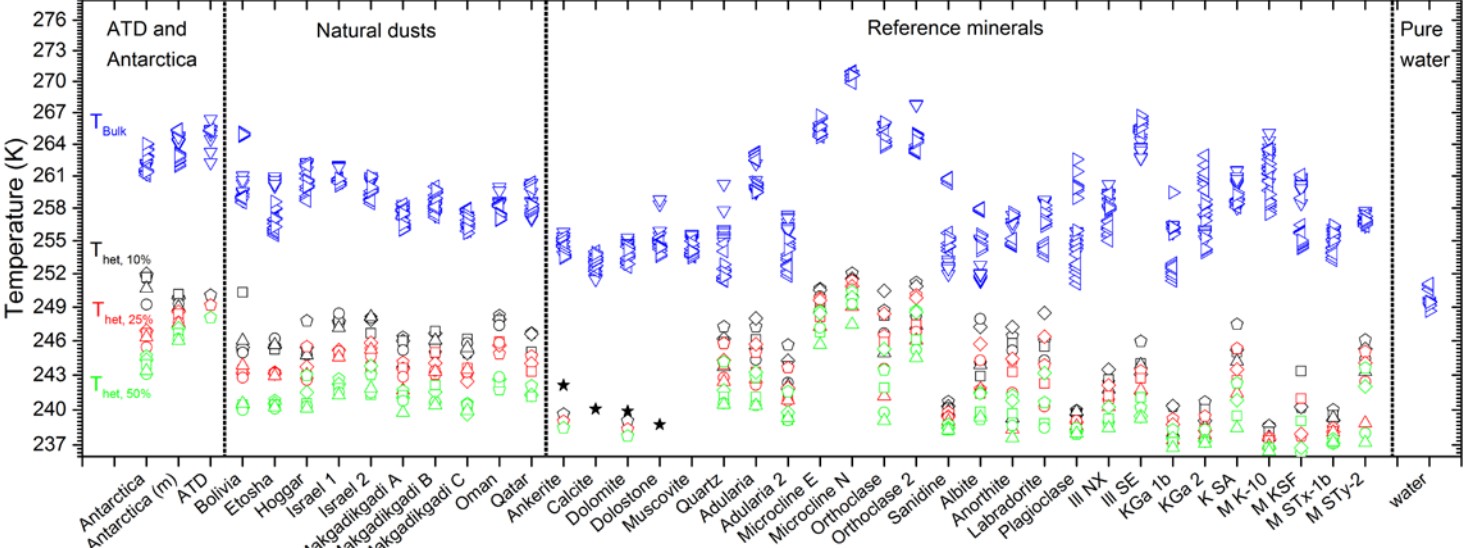

**Figure 4.** Overview of emulsion and bulk freezing temperatures of natural dusts and reference minerals determined from DSC experiments. Reference samples are from this study and from Pinti et al. (2012). Emulsion freezing experiments were carried out with suspension concentrations of 0.5 wt% (triangles), 1 wt% (circles), 2 wt% (squares), 5 wt% (diamonds), and 10 wt% (pentagons). Temperatures of 10% heterogeneously frozen water volume ($T_{het,10\%}$), 25% heterogeneously frozen water volume ($T_{het,25\%}$), and 50% heterogeneously frozen water volume ($T_{het,50\%}$) are given by black, red, and green symbols, respectively. Blue symbols refer to bulk measurements carried out with a 5 wt% suspension. Different symbols refer to different bulk samples. On the far right, bulk measurements of pure water are shown for comparison, with freezing temperatures < 252.5 K. Homogeneous freezing in emulsion samples occurs at T < 236.5 K.





**Figure 5a.** DSC thermograms of frequently found minerals: ankerite, calcite, dolomite, dolostone, muscovite and quartz. Suspension concentrations are given in the legend.





**Figure 5b.** DSC thermograms of K-feldspars: adularia 1 and 2, microcline E, microcline N, orthoclase 1and 2, sanidine. Suspension concentrations are given in the legend.





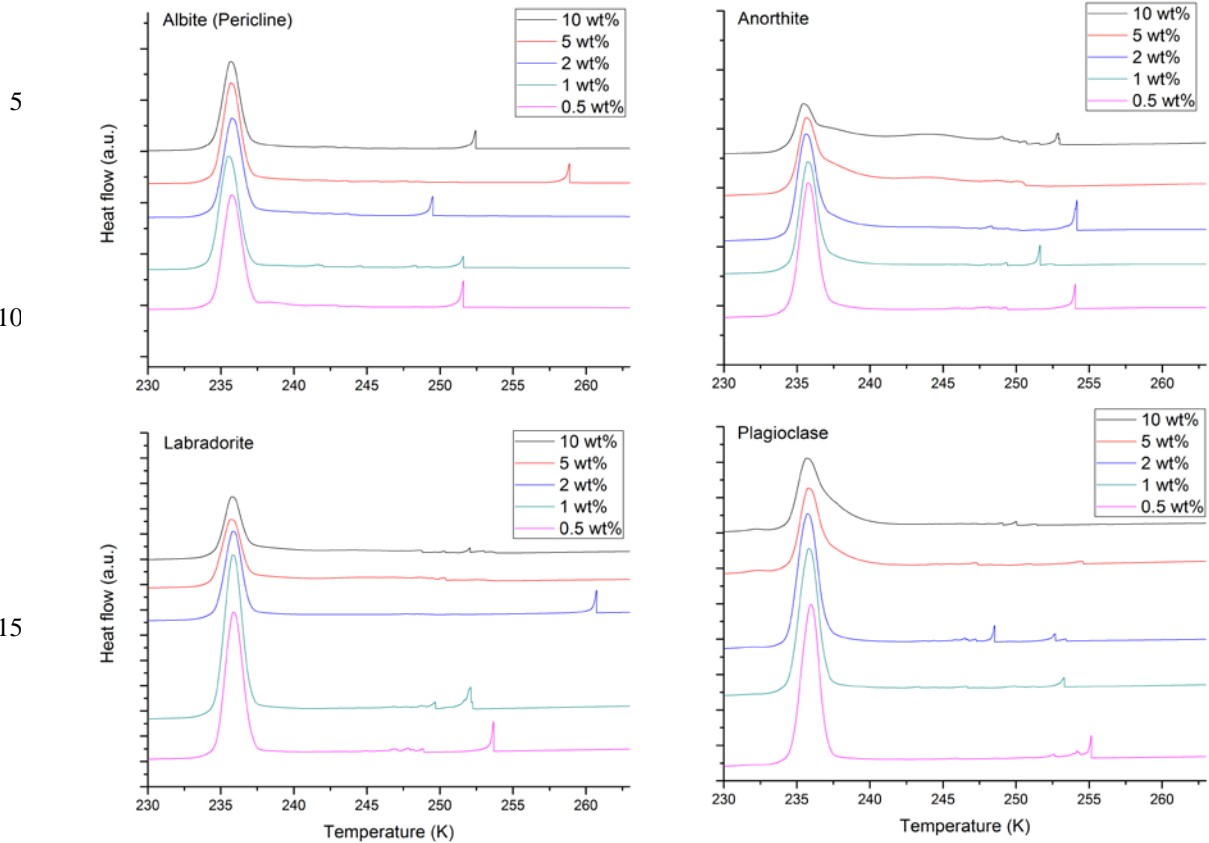

**Figure 5c.** DSC thermograms of (Na, Ca)-feldspars: albite (pericline), anorthite, labradorite, plagioclase. Suspension concentrations are given in the legend.





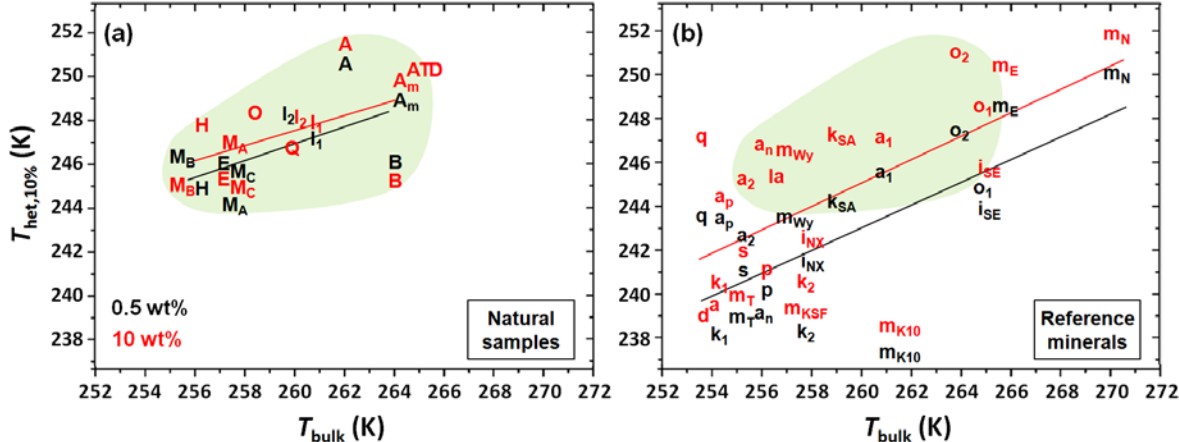

**Figure 6.** Correlation between bulk and emulsion freezing of the natural dust samples (panel a) and reference samples (panel b). Average bulk freezing temperatures are plotted against temperatures of 10% heterogeneously frozen fraction ($T_{het,10\%}$) for suspensions with concentrations of 0.5 wt% (black numbers) and 10 wt% (red numbers). Slopes are $0.0007 \pm 0.0004$ (0.5 wt% suspension) and $0.0006 \pm 0.0003$ (10 wt% suspension) for the natural dust samples (panel a) and $0.0009 \pm 0.0003$ (0.5 wt% and 10 wt% suspensions) for the reference samples (panel b). Sample code for the natural dust samples (panel a): (A) Antarctica, ($A_m$) Antarctica milled, (ATD) Arizona Test Dust, (B) Bolivia, (E) Etosha, (H) Hoggar, ($I_1$, $I_2$) Israel 1 and 2, ($M_A$, $M_B$, $M_C$) Makgadikgadi A, B and C, (O) Oman, (Q) Qatar. Sample code for the reference minerals (panel b): (a) ankerite, (d) dolomite, (q) quartz, ($a_1$, $a_2$) adularia 1 and 2, ($m_E$, $m_N$) microcline E and N, ($o_1$ and $o_2$) orthoclase 1 and 2, (s) sanidine, ($a_p$) albite (pericline), ($a_n$) anorthite, (la) labradorite, (p) plagioclase, ($i_{NX}$, $i_{SE}$) illite NX and SE, ($k_1$, $k_2$, $k_{SA}$) kaolinite KGa-1b, KGa-2 and SA, ($m_{K10}$, $m_{KSF}$, $m_T$, $m_{Wy}$) montmorillonite K-10, KSF, STx-1b and SWy-2. Green shading: typical freezing temperatures of natural samples.