# Peer review of "Ice nucleation efficiency of natural dust samples in the immersion mode"

_Atmospheric Chemistry and Physics, 2016_

## Referee Comment (RC1) · Anonymous Referee #1 · 6 Jun 2016

Review of "Ice nucleation efficiency of natural dust samples in the immersion mode" by Kaufmann et al.

**Summary**
The authors investigated the relationship between the immersion freezing behavior of diverse natural dusts from the ground as well as reference mineral samples and their mineralogical compositions (that are based on XRD). The immersion freezing measurements were conducted using a DSC, in which bulk powders were emulsified and homogenized in a mixture of mineral oil and lanolin. The authors evaluated the immersion freezing behavior in two metrics, $f_{ice}$ (ice nucleation active particle fraction) and $T_{het,xx\%}$ (freezing temperature, $T$, of given frozen fraction ranging from 10 to 50%), in the sub-zero temperatures above ~236.5 K.

      The major finding of this work is that a majority of surface dust samples exhibit similar freezing behavior despite the difference/variation in mineralogy. In turn, the authors suggest the atmospheric dust freezing to be potentially represented in global models in a simple manner (P20 L11-12). Notable exceptions are the samples with microcline that are known as a highly efficient immersion freezing component of dust. Nonetheless, the authors imply that microcline-containing particles are not abundant in the atmosphere and, hence, may have overall small contributions to atmospheric ice nucleation (IN) and glaciation (e.g., P20 L22-25).

**General comments**
The authors conducted very careful and dedicated experimental works. The manuscript is well organized and carefully written to derive a delicate conclusion (i.e., P2 L1-4; P21 L22). The research topic is an important addition to ACP. The authors are knowledgeable in the subject and perceptive about the importance of mineralogy-resolved IN study as they clearly state the necessity of further investigations (e.g., analysis of airborne dust mineralogy and associated modeling simulation works, P20 L25-26; P21 L2-4). I support publication of this manuscript in ACP after the following comments are properly addressed.

**Major comments**
P22 L9-11: The authors suggest that there is only negligible amount of microcline in natural dusts, such that atmospheric IN triggered by microcline may be negligible. The authors did a great job in justifying and documenting their mineralogy results of bulk powder with manual rock interpretations (Sect. 5.2.1). That said, the authors report the XRD diffraction "accuracy" of ± 15% based on the comparison of two XRD analyses for the very same commercial standard material (i.e., ATD) in P10 L11 and Appendix B. Concerning given XRD accuracy, I feel that the statement in P22 L9-11 is a bit too aggressive. As microcline can be K-feldspar (and is not a rare feldspar), we can at least assume it is there in general. The authors may consider softening the tone.

P11 L21-22: Regarding sanidine, the results reported in *Harrison et al.* (2016, ACPD) suggests no difference between sanidine and several microcline samples in terms of IN. If an atmospherically representative sanidine was IN active, the discussion regarding the microcline-scaling IN activity (e.g., P1 L30-31) would be misleading? More or less, I just wonder if it would change any of the authors' conclusions or not.

While the authors report the results of their DSC freezing experiments carried out with the homogenized samples (P4 L10-11), it seems their XRD composition measurements are based on the bulk powder samples without any pretreatment (P5 L24). The size distribution characterization (SMPS/APS as well as EM) may have been performed with the sieved-aerosolized samples (P5 L18-26). Since the authors combine these three independent results afterwards for their data analyses and interpretations, the reviewer suggests the authors to clarify the followings:

- Sample homogenization may do more than just emulsification; e.g., promoting the particle breakup, altering the abundance of certain components, changing the size distribution and

scratching the surface of particles? Any comments? As the authors might be aware, alternations in size and composition, especially for a composite material, are often inherently related (as they discuss some in Sect. 6.4; P19 L5-13).

- Can the authors justify a consistency of the size distributions amongst these individual measurements, especially with respect to aerosolized particles vs. homogenized/emulsified particles? Otherwise, the assumption of the consistency should be clearly stated in text (e.g., P6 L2 and P14 L17-19). I do agree with the authors that sieving with a 32 μm grid helps represent the size distribution of airborne dusts.

- In terms of the particle size, my feeling is as follows; bulk powder > sieved-bulk > aerosolized (EM) > aerosolized (SMPS/APS) > homogenized. It seems that the authors use the aerosolized (SMPS/APS) data as a reference of particle size distributions in homogenized droplets. If so, wouldn't that means the authors may be overestimating $V_p$ and $f_{act}$ in eqn. 3 and eqn. 8, respectively (and underestimating $n$ in eqn. 4)? Concerning aerosolized vs. homogenized, the $f_{act}$ error may be even larger than the values given in Sect. A4? Currently, only EM vs. SMPS/APS is discussed in A1. If that is the case, the overall potential impact should be stated in text.

**Minor comments**

P3 L31: For clarity, the authors may consider rephrasing "natural dust samples" to "surface dust samples"? The authors may consider modifying the title accordingly as well.

P5 L18-20: How did the authors aerosolize the bulk powders? The method (incl. generator spec.) should be briefly described here.

P18 L12-14 & L26-28: What exactly the authors mean for "systematic errors"? I encourage the authors to extend the discussion in a bit more detail. The IN research community seems putting some efforts to tackle the issue regarding data diversity amongst many different techniques recently. The authors may at least cite proper papers.

P20 L6-8: So what is the atmospheric implication of typical IN (that is, emulsion measurement results) vs. best IN (that is, based on bulk)? According to the Appendix A4 (P23 L30-31), using bulk may have some technical issues, correct? This point should be clarified in the main manuscript (e.g., either in Sect. 6.2.2 or Sect. 7).

P20 L10-12: These sentences seem speculative and seem not match with the focus of the current manuscript. Some parts are opinioned. I suggest rephrasing.

**Specific & Technical comments**

P1 L22: best particles/sites → best ice-nucleating particles/sites

P2 L12 and all "IN" hereafter: ice nuclei (IN) → ice-nucleating particles (INPs) according to *Vali et al*. (2015, ACP)?

P3 L7: *Augustin-Bauditz et al*. (2016, ACP doi:10.5194/acp-16-5531-2016) may be a good additional reference regarding the effect of biological materials on mineral dusts in immersion freezing behavior.

P3 L18: *Wang et al*. (2016, Nature Geosci. doi:10.1038/ngeo2705) may be a good ref to add for the composition transfer function from soil to airborne dust.

P3 L20: important → abundant or dominant?

P3 L26: define "large" quantitatively

P3 L32: I disagree. The authors applied a number of mechanical processes. See my major comment. It seems heat and additional mixing may have been applied to a subset of samples (P22 L29)?

P4 L6: → best available ice-nucleating particles/sites

P4 L8-9: I suggest defining the "lower average freezing temperature" here. The authors may consider moving P5 L5-6 to this part.

P5 L24: → …composition of the bulk powder samples was measured by XRD

P6 L8: Reference/explanation for 2.6 g/cm$^3$ is missing.

P9 L10: → number of ice-nucleating particles

P9L14: The authors may explain the usefulness and implication of the $D_{pl}$ parameter here.

P13 L5-6: Please clarify what the authors mean for "minor components". It seems quartz and muscovite are not that IN active according to the results given in Table 5. In general, kaolinite seems containing some K-feldspar (P13 L20-23), which may be responsible for their high IN as inferred in Table 5 as compared to other reference samples. The authors mean it as a minor component?

P19 L29-30: The word "should" is bothering. Any particular references?

P21 L11: → comparable in size after the processing, such as sieving and aerosolization (the authors may consider making a similar statement in P14 L17-19 to clarify this point)

P22 L1: The influence of agglomeration alone on IN should be discussed in Sect. 6.4 with proper citation (e.g., *Emersic et al*., 2016, ACP, and references therein). Otherwise, remove the agglomeration word.

P22 L11: analysis of dust samples → analysis of airborne dust samples

P22 L11: Would the analysis of ice residual particles may help (e.g., *Kupiszewski et al*., 2015, AMT) as a future work?

P22 L16: largest → be more quantitative, put the uncertainty values with respect to $f_{act}$

P37 Table 5: Two different fonts are involved.

P38 Fig. 1: The x-axis should read "droplet diameter"?

P41 Caption: Oman and Qatar → Qatar and Oman

Appendix B: The source of the uncertainty may include the sample itself as well. For instance, ATD is a material composite, and the sample may not be completely homogeneous in terms of mineralogical distribution even within a same batch. The authors may consider briefly mentioning it.

---

## Referee Comment (RC2) · Anonymous Referee #2 · 11 Jun 2016

Review of "Ice nucleation efficiency of natural dust samples in the immersion mode" by Kaufmann et al.

In this study, Kaufmann et al. use a DSC method to examine the nucleating behavior of a wide range of both natural dusts and reference minerals. It is found that the variability in freezing behavior for natural dusts is relatively small. The consequences of this finding is that for model studies, it may be sufficient to represent natural dusts with a single paramaterisation, at least in the temperature ranges examined during this study. The difference in variability in the freezing behavior between natural dust samples and reference materials, which was found to be greater in the case of the latter, is also a key finding, is sure to be of interest to researchers in this area.

My main comments/questions on the paper surround the experimental procedure, and how the data is interpreted. Following clarification of these points, I would recommend the paper for publication in ACP.

Comments and Suggestions:

- It is not immediately obvious why the data from DSC measurements cannot be normalized to nucleation rates or ice active site densities. I can envisage some difficulties in doing this, but a statement on why nucleation rates or ice active site densities are not calculated would be of value to the reader.
- At this point in time, there are two other pertinent papers which are in peer review in ACPD (Harrison et al., 2016; Peckhaus et al., 2016), which are not considered here, but I would highlight that they are very relevant. For the final ACP version of this paper, if these related papers are accepted prior to this one, I would certainly include discussion of them.
- Throughout the paper, IN is used, instead of INPs. I would consider changing this as per (Vali et al., 2015)
- P1L18: for clarity, I would add point out that the 2 um figure given here is from the number distribution.
- P3L7-8 an L18-19: The references here don't all match with the statements made on how organic matter can influence ice nucleating activity, in particular, Baker 2005 and to a lesser extent maybe Hallar 2011; neither of these studies examined ice nucleation as far as I'm aware. Also, there are multiple more pertinent references here e.g. (Augustin-Bauditz et al., 2016; O'Sullivan et al., 2014; Tobo et al., 2014)
- P3L20: "important" is a very qualitative word- I suggest changing to something more concise.
- P4 experimental setup: Very high concentrations of dusts are used during some experiments, up to 50% (!). The authors refer to these as suspensions (by 50 wt %, I envisage this is more of a slurry than a "suspension"), but no indication is given on their stability. Emersic et al. (2015) suggest that aggregation, and surface area occlusion in droplets of 1 wt% is an issue for droplet freezing experiments- could this be an issue

for these experiments at much higher concentrations? A discussion on these points is warranted, perhaps in the experimental section.

- P5L 8-17 and Appendix A2: I have missed it elsewhere, but it would be useful to know here how many separate emulsions were examined in the determination of the droplet size distributions, and the total number of droplets examined. Also, this info should be added to the caption of figure 5.

- P5L20: were these wet or dry sieved?

- P6, section 3: If I understand correctly here, the authors are using size distributions measured by SMPS/APS, but are then using this information to estimate the number of particles in suspension droplets. The particle size distributions will be different in the suspension than from the aerosol phase due to aggregation. Will this not lead to significant errors in the calculation of the number of dust particles per droplet, and hence $f_{act}$?

- P19 L14-30: The authors attempt to explain the freezing behaviors of dusts which did not entirely fit with their hypothesis that mineralogical composition is the dominant factor accounting for this. Again, it would seem to me that recent papers in open discussion (Harrison et al., 2016; Peckhaus et al., 2016) are particularly pertinent to the discussion here.

- P19L16-18. Do the authors have data to substantiate that in solution, the milled reference samples do not aggregate also?

- P19L29-30: Perhaps the amount of organic matter could be expected to be small, but the OM content of the dusts was not investigated here. Even trace amounts of organic matter could affect the nucleating abilities of the dusts. Either the authors should further add to arguments that the amounts of OM are too small to affect the freezing behavior, or drop this last sentence.

- P20L12: This relates to my first comment above again: it would be useful to state why the thermogram data cannot be transformed into a parameterization which could be implemented in models.

References:

Augustin-Bauditz, S., Wex, H., Denjean, C., Hartmann, S., Schneider, J., Schmidt, S., Ebert, M. and Stratmann, F.: Laboratory-generated mixtures of mineral dust particles with biological substances: characterization of the particle mixing state and immersion freezing behavior, Atmos. Chem. Phys., 16(9), 5531–5543, doi:10.5194/acp-16-5531-2016, 2016.

Emersic, C., Connolly, P. J., Boult, S., Campana, M. and Li, Z.: Investigating the discrepancy between wet-suspension-and dry-dispersion-derived ice nucleation efficiency of mineral particles, Atmos. Chem. Phys., 15(19), 11311–11326, doi:10.5194/acp-15-11311-2015, 2015.

Harrison, A. D., Whale, T. F., Carpenter, M. A. ., Holden, M. A., Neve, L., O'Sullivan, D., Vergara Temprado, J. and Murray, B. J.: Not all feldspar is equal: a survey of ice nucleating properties across the feldspar group of minerals, Atmos. Chem. Phys. Discuss., (February), 1–26, doi:10.5194/acp-2016-136, 2016.

O'Sullivan, D., Murray, B. J., Malkin, T. L., Whale, T. F., Umo, N. S., Atkinson, J. D., Price, H. C., Baustian, K. J., Browse, J. and Webb, M. E.: Ice nucleation by fertile soil dusts: relative importance of mineral and biogenic components, Atmos. Chem. Phys., 14(4), 1853–1867, doi:10.5194/acp-14-1853-2014, 2014.

Peckhaus, A., Kiselev, A., Hiron, T., Ebert, M. and Leisner, T.: A comparative study of K-rich and Na/Ca-rich feldspar ice nucleating particles in a nanoliter droplet freezing assay, Atmos. Chem. Phys. Discuss., 0, 1–43, doi:10.5194/acp-2016-72, 2016.

Tobo, Y., Demott, P. J., Hill, T. C. J., Prenni, A. J., Swoboda-Colberg, N. G., Franc, G. D. and Kreidenweis, S. M.: Organic matter matters for ice nuclei of agricultural soil origin, Atmos. Chem. Phys., 14(16), 8521–8531, doi:10.5194/acp-14-8521-2014, 2014.

Vali, G., DeMott, P. J., M??hler, O. and Whale, T. F.: Technical Note: A proposal for ice nucleation terminology, Atmos. Chem. Phys., 15(18), 10263–10270, doi:10.5194/acp-15-10263-2015, 2015.

---

## Short Comment (SC1) · 13 Jul 2016

It is surprising to see this statement and highlight of the paper listed in the abstract, that "...global models, in a first approximation, may represent mineral dust as a single species with respect to ice nucleation activity," without reference to a paper published just last year in Atmospheric Chemistry and Physics. The authors of the present paper do a fine job of justifying this point in great detail through laboratory studies, but in the abstract of DeMott et al. (2015) is the following statement: "These studies support the utility of laboratory measurements to obtain atmospherically relevant data on the ice nucleation properties of dust and other particle types, and suggest the suitability of considering all mineral dust as a single type of ice nucleating particle as a useful first-order approximation in numerical modeling investigations." It is hard to read that recommendation as any different than the one made in the present study, so it seems

appropriate to reference this prior work, especially with the similar focus on immersion freezing nucleation.

DeMott, P. J., Prenni, A. J., McMeeking, G. R., Tobo, Y., Sullivan, R. C., Petters, M. D., Niemand, M., Möhler, O., and Kreidenweis, S. M.: Integrating laboratory and field data to quantify the immersion freezing ice nucleation activity of mineral dust particles, Atmos. Chem. Phys., 15, 393–409, doi:10.5194/acp-15-393-2015, 2015.
* * *

---

## Author Comment (AC1) · 23 Aug 2016

*Responses to Reviewer #1*
*We thank the reviewer for the careful reading of the manuscript and the suggestions for improvement. Our point by point responses are given below in italic.*

**Summary**
The authors investigated the relationship between the immersion freezing behavior of diverse natural dusts from the ground as well as reference mineral samples and their mineralogical compositions (that are based on XRD). The immersion freezing measurements were conducted using a DSC, in which bulk powders were emulsified and homogenized in a mixture of mineral oil and lanolin. The authors evaluated the immersion freezing behavior in two metrics, $f_{ice}$ (ice nucleation active particle fraction) and $T_{het,xx\%}$ (freezing temperature, $T$, of given frozen fraction ranging from 10 to 50%), in the sub-zero temperatures above ~236.5 K.

The major finding of this work is that a majority of surface dust samples exhibit similar freezing behavior despite the difference/variation in mineralogy. In turn, the authors suggest the atmospheric dust freezing to be potentially represented in global models in a simple manner (P20 L11-12). Notable exceptions are the samples with microcline that are known as a highly efficient immersion freezing component of dust. Nonetheless, the authors imply that microcline-containing particles are not abundant in the atmosphere and, hence, may have overall small contributions to atmospheric ice nucleation (IN) and glaciation (e.g., P20 L22-25).

**General comments**
The authors conducted very careful and dedicated experimental works. The manuscript is well organized and carefully written to derive a delicate conclusion (i.e., P2 L1-4; P21 L22). The research topic is an important addition to ACP. The authors are knowledgeable in the subject and perceptive about the importance of mineralogy-resolved IN study as they clearly state the necessity of further investigations (e.g., analysis of airborne dust mineralogy and associated modeling simulation works, P20 L25-26; P21 L2-4). I support publication of this manuscript in ACP after the following comments are properly addressed.

**Major comments**
P22 L9-11: The authors suggest that there is only negligible amount of microcline in natural dusts, such that atmospheric IN triggered by microcline may be negligible. The authors did a great job in justifying and documenting their mineralogy results of bulk powder with manual rock interpretations (Sect. 5.2.1). That said, the authors report the XRD diffraction "accuracy" of ± 15% based on the comparison of two XRD analyses for the very same commercial standard material (i.e., ATD) in P10 L11 and Appendix B. Concerning given XRD accuracy, I feel that the statement in P22 L9-11 is a bit too aggressive. As microcline can be K-feldspar (and is not a rare feldspar), we can at least assume it is there in general. The authors may consider softening the tone.
*Our conclusion for microcline was just based on the results from dust samples of this study. Very recently, a study by Boose et al. appeared in ACPD. They found microcline in several samples. We will cite this work in the ACP version and rephrase the text accordingly:*
*"Boose et al. (2016) found microcline present in one out of four investigated airborne dust samples originating from the Sahara and in three out of eight surface-collected dust samples. If microcline particles are indeed common in atmospheric dusts they could be relevant for cloud glaciation at temperatures above 260 K."*

P11 L21-22: Regarding sanidine, the results reported in *Harrison et al.* (2016, ACPD) suggest no difference between sanidine and several microcline samples in terms of IN. If an atmospherically representative sanidine was IN active, the discussion regarding the microcline-scaling IN activity (e.g., P1 L30-31) would be misleading? More or less, I just wonder if it would change any of the authors' conclusions or not.
*It does not change our conclusions because Harrison et al. performed their experiments with microliter*

*droplets containing many particles while in the emulsion droplets, there are only one or a few. We refer to this in the revised manuscript by adding the following text:*

*"Harrison et al. (2016) have recently performed freezing experiments with microliter droplets of aqueous suspensions of a ground sanidine sample. The observed freezing temperatures indicated a similarly high ice-nucleation activity for this sanidine sample as for microclines. The mineralogical composition was investigated by Rietveld refinement of powder XRD patterns confirming sanidine as the dominant feldspar phase, however, without specifying minor components. Considering the high number of particles present in the microliter droplets, the results are not directly applicable to the freezing of emulsion droplets containing only one or a few particles. Nevertheless, we note a quite large variability in ice nucleation activity between sanidine samples, which does not seem to correlate with the mineralogical composition. Based on the emulsion experiments with DSC, we consider sanidine as ice nucleation active but only at lower temperatures compared with microcline."*

While the authors report the results of their DSC freezing experiments carried out with the homogenized samples (P4 L10-11), it seems their XRD composition measurements are based on the bulk powder samples without any pretreatment (P5 L24). The size distribution characterization (SMPS/APS as well as EM) may have been performed with the sieved-aerosolized samples (P5 L18-26).

*The homogenizer was used to produce the emulsions, not to homogenize the dusts. The vigorous stirring of the homogenizer might reduce aggregation. The XRD measurements were performed with the sieved samples. Freezing experiments, size distribution characterization and XRD measurements have all been performed with the sieved samples.*

Since the authors combine these three independent results afterwards for their data analyses and interpretations, the reviewer suggests the authors to clarify the followings:

- Sample homogenization may do more than just emulsfication; e.g., promoting the particle breakup, altering the abundance of certain components, changing the size distribution and scratching the surface of particles? Any comments? As the authors might be aware, alternations in size and composition, especially for a composite material, are often inherently related (as they discuss some in Sect. 6.4; P19 L5-13).

  *We acknowledge the uncertainty in the size distribution as one major uncertainty in the calculation of the fraction of active particles (Appendix A and especially section A1). We think that surface modifications due to scratching with the homogenizer might be important when applied to dry particles rather than particles mixed with oil and water.*

  *We discuss the influence of aggregation in the revised manuscript in Sect. 6.3 by adding the text:*

  *"Ideally, the derived active particle fractions should be independent of suspension concentration. If particles aggregated in suspension, the active particle fraction would be underestimated because the effective number of empty droplets would be larger than the one determined from the size distribution of the dry aerosol. Stronger aggregation is expected at higher concentration leading to an increasing low bias with increasing suspension concentration. To elucidate whether such a tendency is present, the ratios of $f_{act}$ of 0.5 wt% and 2 wt% suspensions, $f_{act}(0.5)/f_{act}(2)$, and the ratio of $f_{act}$ of 2 wt% and 10 wt% suspensions, $f_{act}(2)/f_{act}(10)$, are also listed in Tables 2 and 5. For most natural dust samples the ratios are > 1, indicating some aggregation. The reference samples give a less clear picture. The ratios show quite a large scatter with values between 0.5 and 2 and a tendency to values > 1, indicating aggregation in some cases. Pinti et al. (2012) have discussed the possibility of aggregation for clay minerals concluding that kaolinites show quite strong aggregation mainly at low pH, no aggregation is expected for montmorillonites while no clear information could be obtained for illites. Emersic et al. (2015) hypothesized a possible influence of coagulation to explain the discrepancy between wet-suspension- and dry-dispersion-derived ice nucleation efficiency of mineral particles using kaolinite, NX-illite and a K-feldspar as examples. They showed aggregation for kaolinite using dynamic light-scattering but did not present corresponding data for illite and the K-feldspar."*

- Can the authors justify a consistency of the size distributions amongst these individual measurements, especially with respect to aerosolized particles vs. homogenized/emulsified particles? Otherwise, the assumption of the consistency should be clearly stated in text (e.g., P6 L2 and P14 L17-19). I do agree with the authors that sieving with a 32 µm grid helps represent the size distribution of airborne dusts.
  *Possible inconsistencies are discussed in Appendix A. We extend this discussion in the revised version and write more about aggregation (see above).*

- In terms of the particle size, my feeling is as follows; bulk powder > sieved-bulk > aerosolized (EM) > aerosolized (SMPS/APS) > homogenized. It seems that the authors use the aerosolized (SMPS/APS) data as a reference of particle size distributions in homogenized droplets. If so, wouldn't that means the authors may be overestimating $V_p$ and $f_{act}$ in eqn. 3 and eqn. 8, respectively (and underestimating $n$ in eqn. 4)? Concerning aerosolized vs. homogenized, the $f_{act}$ error may be even larger than the values given in Sect. A4? Currently, only EM vs. SMPS/APS is discussed in A1. If that is the case, the overall potential impact should be stated in text.
  *We add a discussion of aggregation to the revised manuscript where we compare $f_{act}$ obtained for different suspension concentrations, indicating a tendency of aggregation in the emulsion droplets (see above). Particles in the submicron size range are more frequently primary particles than aggregates. The size distribution from SMPS/APS peaks in the submicron region, therefore we think that primary particles dominate. In case of primary particles, the emulsification by the homogenizer should therefore not lead to a further breakup of particles.*

**Minor comments**

P3 L31: For clarity, the authors may consider rephrasing "natural dust samples" to "surface dust samples"? The authors may consider modifying the title accordingly as well.
*We prefer to keep "natural dust samples", because we think that with the applied pretreatment of only sieving, our surface-collected samples should represent well airborne samples.*

P5 L18-20: How did the authors aerosolize the bulk powders? The method (incl. generator spec.) should be briefly described here.
*The bulk powders were aerosolized in a Fluidized Bed Aerosol Generator (TSI Model 3400A). We add this information to the revised manuscript*

P18 L12-14 & L26-28: What exactly the authors mean for "systematic errors"? I encourage the authors to extend the discussion in a bit more detail. The IN research community seems putting some efforts to tackle the issue regarding data diversity amongst many different techniques recently. The authors may at least cite proper papers.
*We discriminate between random errors, which decrease by averaging repeated measurements and systematic errors, which remain, even if repeated measurements are averaged. Systematic errors need to be quantified by measuring a reference (calibration) sample with known measurement value or by comparing with a reference measurement performed with another instrument/technique. Unfortunately, in atmospheric sciences, such reference samples or reference techniques are usually not available and the systematic errors can only be guessed.*

P20 L6-8: So what is the atmospheric implication of typical IN (that is, emulsion measurement results) vs. best IN (that is, based on bulk)? According to the Appendix A4 (P23 L30-31), using bulk may have some technical issues, correct? This point should be clarified in the main manuscript (e.g., either in Sect. 6.2.2 or Sect. 7).
*Emulsion measurements give an information about the most probable freezing behavior/temperature. Bulk measurements give information about the "best" possible freezing behavior/temperature. Bulk temperatures are usually much higher than emulsion temperatures, but only few INPs are able to induce freezing at temperatures observed for the bulk measurements. To make this clearer, we add the following sentence to*

*section 6.2.2: "Comparison with freezing temperatures observed for emulsion samples show that experiments with suspensions containing a high number of particles do not represent the freezing behaviour of typical INPs in a sample."*

P20 L10-12: These sentences seem speculative and seem not match with the focus of the current manuscript. Some parts are opinionated. I suggest rephrasing.
*We deleted this sentence in the revised manuscript.*

**Specific & Technical comments**
P1 L22: best particles/sites → best ice-nucleating particles/sites
*Done.*

P2 L12 and all "IN" hereafter: ice nuclei (IN) → ice-nucleating particles (INPs) according to *Vali et al.* (2015, ACP)?
*We use INP in the revised manuscript.*

P3 L7: *Augustin-Bauditz et al.* (2016, ACP doi:10.5194/acp-16-5531-2016) may be a good additional reference regarding the effect of biological materials on mineral dusts in immersion freezing behavior.
*Thank you for pointing out this paper, we refer to it in the revised manuscript.*

P3 L18: *Wang et al.* (2016, Nature Geosci. doi:10.1038/ngeo2705) may be a good ref to add for the composition transfer function from soil to airborne dust.
*Thank you for pointing out this paper, we refer to it in the revised manuscript.*

P3 L20: important → abundant or dominant?
*We changed to "abundant"*

P3 L26: define "large" quantitatively
*We cite here the abstract of Atkinson et al. (2013). It was not more specific.*

P3 L32: I disagree. The authors applied a number of mechanical processes. See my major comment. It seems heat and additional mixing may have been applied to a subset of samples (P22 L29)?
*The only pretreatment of the mineral dust samples was sieving. P22 L29 refers to the mineral oil / lanolin mixture. This mixture does not include mineral dust particles. The mineral dust was added to the water. Subsequently, the oil and the aqueous phase were mixed together by vigorous stirring with a homogenizer leading to the emulsion.*

P4 L6: → best available ice-nucleating particles/sites
*Done.*

P4 L8-9: I suggest defining the "lower average freezing temperature" here. The authors may consider moving P5 L5-6 to this part.
*We reversed the order and now explain first bulk freezing measurements and then emulsion freezing experiments.*

P5 L24: → …composition of the bulk powder samples was measured by XRD
*We rephrased: "The mineralogical composition of the sieved natural dust samples and the milled reference samples was measured by X-ray diffraction (XRD).*

P6 L8: Reference/explanation for 2.6 g/cm$^3$ is missing.
*This value is taken from Möhler et al. (2006). We add this reference to the revised version.*

P9 L10: → number of ice-nucleating particles
*We changed to "INPs"*

P9L14: The authors may explain the usefulness and implication of the $D_{pl}$ parameter here.
*We explain this parameter better in the revised manuscript.*

P13 L5-6: Please clarify what the authors mean for "minor components". It seems quartz and muscovite are not that IN active according to the results given in Table 5. In general, kaolinite seems containing some K-feldspar (P13 L20-23), which may be responsible for their high IN as inferred in Table 5 as compared to other reference samples. The authors mean it as a minor component?
*Indeed, the minor components can explain the ice-nucleation activity of the anorthite sample only partly. We therefore rephrase: "For the anorthite sample, in addition to plagioclase as the main component (59%), 12% quartz, 8% muscovite and 5% kaolinite were identified as most abundant minor components. These minor components can explain the ice nucleation activity reaching to a higher temperature compared with the other (Na,Ca)-feldspar samples only partly."*

P19 L29-30: The word "should" is bothering. Any particular references?
*Following reviewer #2, we deleted this sentence because it is speculative.*

P21 L11: → comparable in size after the processing, such as sieving and aerosolization (the authors may consider making a similar statement in P14 L17-19 to clarify this point).
*We add the following sentence on P14 L19: "We therefore consider the sieved ground-collected natural dust samples as comparable in size with airborne mineral dusts."*

P22 L1: The influence of agglomeration alone on IN should be discussed in Sect. 6.4 with proper citation (e.g., *Emersic et al.*, 2016, ACP, and references therein). Otherwise, remove the agglomeration word.
*We add a discussion of agglomeration at the beginning of Sect. 6.3:*
*"Depending on size and suspension concentration, droplets of the investigated emulsions may be empty or contain one or a few particles. Empty droplets as well as droplets containing only ice nucleation inactive particles contribute to the homogeneous freezing signal in the DSC curves. Tables 2 and 5 list in the second column $D_{p1}$, the average diameter of a droplet with 1 particle inside for 2 wt% suspensions, indicating that smaller particles are empty and larger ones contain one or a few particles. Assuming that all particles are able to nucleate ice, the heterogeneously frozen water volume fraction, $p_{het}$, can be calculated and compared with the measured one, $p_{het,lab}$. The ice-nucleation active particle fractions were calculated for all concentrations and are given in Tables 2 and 5 for the 2 wt% suspensions. They range from $f_{act} = 0.025 – 0.32$ (Table 2) for the natural dust samples excluding ATD and from $f_{act} = 0.0004 – 0.64$ for the reference minerals (Table 5). Ideally, the derived active particle fractions should be independent of suspension concentration. If particles aggregated in suspension, the active particle fraction would be underestimated because the effective number of empty droplets would be larger than the one determined from the size distribution of the dry aerosol. Stronger aggregation is expected at higher concentration leading to an increasing low bias with increasing suspension concentration. To elucidate whether such a tendency is present, the ratios of $f_{act}$ of 0.5 wt% and 2 wt% suspensions, $f_{act}(0.5)/f_{act}(2)$, and the ratio of $f_{act}$ of 2 wt% and 10 wt% suspensions, $f_{act}(2)/f_{act}(10)$, are also listed in Tables 2 and 5. For most natural dust samples the ratios are > 1, indicating some aggregation. The reference samples give a less clear picture. The ratios show quite a large scatter with values between 0.5 and 2 and a tendency to values > 1, indicating aggregation in some cases. Pinti et al. (2012) have discussed the possibility of aggregation for clay minerals concluding that kaolinites show quite strong aggregation mainly at low pH, no aggregation is expected for montmorillonites while no clear information could be obtained for illites. Emersic et al. (2015) hypothesized a possible influence of coagulation to explain the discrepancy between wet-suspension- and dry-dispersion-derived ice nucleation efficiency of mineral particles using kaolinite, NX-illite and a K-*

*feldspar as examples. They showed aggregation for kaolinite using dynamic light-scattering but did not present corresponding data for illite and the K-feldspar."*

P22 L11: analysis of dust samples → analysis of airborne dust samples
*Done.*

P22 L11: Would the analysis of ice residual particles may help (e.g., *Kupiszewski et al.*, 2015, AMT) as a future work?
*EDX measurements on single particles may indeed help and are a good idea for future work.*

P22 L16: largest → be more quantitative, put the uncertainty values with respect to $f_{act}$
*"largest" is specified in the following text as an uncertainty by a factor 2.8 – 4.4.*

P37 Table 5: Two different fonts are involved.
*We correct this in the revised manuscript.*

P38 Fig. 1: The x-axis should read "droplet diameter"?
*Yes, indeed. Thank you for pointing this out.*

P41 Caption: Oman and Qatar → Qatar and Oman
*Corrected.*

Appendix B: The source of the uncertainty may include the sample itself as well. For instance, ATD is a material composite, and the sample may not be completely homogeneous in terms of mineralogical distribution even within a same batch. The authors may consider briefly mentioning it.
*This is a good point. We add the following sentence to Appendix B: "ATD is a material composite. There might be variations in exact composition between batches or even within a batch. This might be a reason for discrepancies in addition to the accuracy of the XRD evaluation."*

---

## Author Comment (AC2) · 23 Aug 2016

**Responses to reviewer #2**

*We thank the reviewer for the thoughtful comments. Our responses are given in italic below.*

In this study, Kaufmann et al. use a DSC method to examine the nucleating behavior of a wide range of both natural dusts and reference minerals. It is found that the variability in freezing behavior for natural dusts is relatively small. The consequences of this finding is that for model studies, it may be sufficient to represent natural dusts with a single parameterisation, at least in the temperature ranges examined during this study. The difference in variability in the freezing behavior between natural dust samples and reference materials, which was found to be greater in the case of the latter, is also a key finding, is sure to be of interest to researchers in this area.

My main comments/questions on the paper surround the experimental procedure, and how the data is interpreted. Following clarification of these points, I would recommend the paper for publication in ACP.

Comments and Suggestions:

- It is not immediately obvious why the data from DSC measurements cannot be normalized to nucleation rates or ice active site densities. I can envisage some difficulties in doing this, but a statement on why nucleation rates or ice active site densities are not calculated would be of value to the reader.
  *In DSC measurements the heat flow due to freezing is measured. The presented data was for emulsion measurements where a high number of droplets with different sizes and containing different numbers of particles freeze. Nucleation rates are not a direct result of the experiments. To derive nucleation rates and ice active site densities, a complete modeling of the DSC curves is needed as was done in Marcolli et al. (2007) for Arizona test dust. Such a modeling is a major task and was therefore not intended in this study, which focuses on the mineralogical composition. It could be the subject of a follow-up study.*

- At this point in time, there are two other pertinent papers which are in peer review in ACPD (Harrison et al., 2016; Peckhaus et al., 2016), which are not considered here, but I would highlight that they are very relevant. For the final ACP version of this paper, if these related papers are accepted prior to this one, I would certainly include discussion of them.
  *Thank you for pointing them out. We discuss the aspects of these paper that are relevant for our study in the ACP version:*
  *In Sect. 5.2.2: "Harrison et al. (2016) have recently performed freezing experiments with microliter droplets of aqueous suspensions of a ground sanidine sample. The observed freezing temperatures indicated a similarly high ice nucleation activity for this sanidine sample as for microclines. The mineralogical composition was investigated by Rietveld refinement of powder XRD patterns confirming sanidine as the dominant feldspar phase, however, without specifying minor components. Considering the high number of particles present in the microliter droplets, the results are not directly applicable to the freezing of emulsion droplets containing only one or a few particles. Nevertheless, we note a quite large variability in ice nucleation activity between sanidine samples, which does not seem to correlate with the mineralogical composition. Based on the emulsion experiments with DSC, we consider sanidine as ice*

*nucleation active but only at lower temperatures compared with microcline.”*
*And also in Sect. 5.2.2: “Harrison et al. (2016) investigated three albite samples with very different ice nucleation activities. One of them showed similarly high freezing temperatures as the microcline samples but lost its activity over time while suspended in water. The other samples showed distinctly lower freezing temperatures than the microclines but slightly higher ones than the plagioclase samples that they also investigated.”*
*In Sect. 6.4: “Peckhaus et al. (2016) investigated four milled feldspar minerals in freezing experiments. Bulk mineralogical composition determined by XRD revealed the three K-rich samples to consist mainly of microcline (76 – 80 %) with a minor (Na, Ca)-feldspar component (16 – 24 %). They used an environmental scanning electron microscope to record images of single particles and energy dispersive X-ray (EDX) to infer the mineralogical composition of the particles. The SEM images showed agglomerates consisting of several large particles with smaller particle fragments on their surface. The (Na,Ca)-feldspar sample exhibited large inter-particle variability in Na/Ca ratio. The K-rich feldspar particles contained varying amounts of sodium and also calcium. Only one K-rich feldspar contained some pure K-feldspar particles with no share of sodium. Interestingly, this sample showed the highest freezing temperatures.”*

- Throughout the paper, IN is used, instead of INPs. I would consider changing this as per (Vali et al., 2015)
  *We use INP in the revised manuscript.*

- P1L18: for clarity, I would add point out that the 2 um figure given here is from the number distribution.
  *We reformulated: “For emulsion measurements, water droplets with a size distribution peaking at about 2μm,”*

- P3L7-8 an L18-19: The references here don't all match with the statements made on how organic matter can influence ice nucleating activity, in particular, Baker 2005 and to a lesser extent maybe Hallar 2011; neither of these studies examined ice nucleation as far as I'm aware. Also, there are multiple more pertinent references here e.g. (Augustin-Bauditz et al., 2016; O'Sullivan et al., 2014; Tobo et al., 2014)
  *We removed references that were not fitting and added the suggested references.*

- P3L20: “important” is a very qualitative word- I suggest changing to something more concise.
  *We replaced “important” by “abundant”.*

- P4 experimental setup: Very high concentrations of dusts are used during some experiments, up to 50% (!). The authors refer to these as suspensions (by 50 wt %, I envisage this is more of a slurry than a “suspension”), but no indication is given on their stability. Emersic et al. (2015) suggest that aggregation, and surface area occlusion in droplets of 1 wt% is an issue for droplet freezing experiments- could this be an issue for these experiments at much higher concentrations? A discussion on these points is warranted, perhaps in the experimental section.

*50 wt% is indeed a very high concentrations and was only used in cases where the DSC signal was too low to be detectable at a lower concentration. We add a discussion of aggregation to Sect. 6.3 of the revised manuscript:*

*"Depending on size and suspension concentration, droplets of the investigated emulsions may be empty or contain one or a few particles. Empty droplets as well as droplets containing only ice nucleation inactive particles contribute to the homogeneous freezing signal in the DSC curves. Tables 2 and 5 list in the second column $D_{p1}$, the average diameter of a droplet with 1 particle inside for 2 wt% suspensions, indicating that smaller particles are empty and larger ones contain one or a few particles. Assuming that all particles are able to nucleate ice, the heterogeneously frozen water volume fraction, $p_{het}$, can be calculated and compared with the measured one, $p_{het,lab}$. The ice nucleation active particle fractions were calculated for all concentrations and are given in Tables 2 and 5 for the 2 wt% suspensions. They range from $f_{act} = 0.025 – 0.32$ (Table 2) for the natural dust samples excluding ATD and from $f_{act} = 0.0004 – 0.64$ for the reference minerals (Table 5). Ideally, the derived active particle fractions should be independent of suspension concentration. If particles aggregated in suspension, the active particle fraction would be underestimated because the effective number of empty droplets would be larger than the one determined from the size distribution of the dry aerosol. Stronger aggregation is expected at higher concentration leading to an increasing low bias with increasing suspension concentration. To elucidate whether such a tendency is present, the ratios of $f_{act}$ of 0.5 wt% and 2 wt% suspensions, $f_{act}(0.5)/f_{act}(2)$, and the ratio of $f_{act}$ of 2 wt% and 10 wt% suspensions, $f_{act}(2)/f_{act}(10)$, are also listed in Tables 2 and 5. For most natural dust samples the ratios are > 1, indicating some aggregation. The reference samples give a less clear picture. The ratios show quite a large scatter with values between 0.5 and 2 and a tendency to values > 1, indicating some aggregation. Pinti et al. (2012) have discussed the possibility of aggregation for clay minerals concluding that kaolinites show quite strong aggregation mainly at low pH, no aggregation is expected for montmorillonites while no clear information could be obtained for illites. Emersic et al. (2015) hypothesized a possible influence of coagulation to explain the discrepancy between wet-suspension- and dry-dispersion-derived ice nucleation efficiency of mineral particles using kaolinite, NX-illite and a K-feldspar as examples. They showed aggregation for kaolinite using dynamic light-scattering but did not present corresponding data for illite and the K-feldspar."*

- P5L 8-17 and Appendix A2: I have missed it elsewhere, but it would be useful to know here how many separate emulsions were examined in the determination of the droplet size distributions, and the total number of droplets examined. Also, this info should be added to the caption of figure 5.
  *We add this information in the revised manuscript on page 5.*

- P5L20: were these wet or dry sieved?
  *The dry samples were sieved. We add this information by adding the sentence:*
  *"Sieving was performed with the dry samples."*

- P6, section 3: If I understand correctly here, the authors are using size distributions measured by SMPS/APS, but are then using this information to estimate the number of particles in suspension droplets. The particle size distributions will be different in the

suspension than from the aerosol phase due to aggregation. Will this not lead to significant errors in the calculation of the number of dust particles per droplet, and hence $f_{act}$?

*We discussed uncertainties of the fraction of active INPs in Appendix A and concluded that a major uncertainty is indeed due to the uncertainties related to the size distribution. In the revised manuscript we discuss also the effect of aggregation (see response above).*

- P19 L14-30: The authors attempt to explain the freezing behaviors of dusts which did not entirely fit with their hypothesis that mineralogical composition is the dominant factor accounting for this. Again, it would seem to me that recent papers in open discussion (Harrison et al., 2016; Peckhaus et al., 2016) are particularly pertinent to the discussion here.

*We add a discussion of these two paper to the revised manuscript, see the response to the comment above.*

P19L16-18. Do the authors have data to substantiate that in solution, the milled reference samples do not aggregate also?
*We added this discussion to the revised manuscript. See above.*

- P19L29-30: Perhaps the amount of organic matter could be expected to be small, but the OM content of the dusts was not investigated here. Even trace amounts of organic matter could affect the nucleating abilities of the dusts. Either the authors should further add to arguments that the amounts of OM are too small to affect the freezing behavior, or drop this last sentence.
*Indeed, little is known about the content of organic and biological matter and therefore we drop the sentence.*

- P20L12: This relates to my first comment above again: it would be useful to state why the thermogram data cannot be transformed into a parameterization which could be implemented in models.
*See the response above.*

References:

Augustin-Bauditz, S., Wex, H., Denjean, C., Hartmann, S., Schneider, J., Schmidt, S., Ebert, M. and Stratmann, F.: Laboratory-generated mixtures of mineral dust particles with biological substances: characterization of the particle mixing state and immersion freezing behavior, Atmos. Chem. Phys., 16(9), 5531–5543, doi:10.5194/acp-16-5531-2016, 2016.

Emersic, C., Connolly, P. J., Boult, S., Campana, M. and Li, Z.: Investigating the discrepancy between wet-suspension-and dry-dispersion-derived ice nucleation efficiency of mineral particles, Atmos. Chem. Phys., 15(19), 11311–11326, doi:10.5194/acp-15-11311-2015, 2015.

Harrison, A. D., Whale, T. F., Carpenter, M. A. ., Holden, M. A., Neve, L., O'Sullivan, D., Vergara Temprado, J. and Murray, B. J.: Not all feldspar is equal: a survey of ice nucleating properties across the feldspar group of minerals, Atmos. Chem. Phys. Discuss., (February), 1–26, doi:10.5194/acp-2016-136, 2016.

O'Sullivan, D., Murray, B. J., Malkin, T. L., Whale, T. F., Umo, N. S., Atkinson, J. D., Price, H. C., Baustian, K. J., Browse, J. and Webb, M. E.: Ice nucleation by fertile soil dusts: relative importance of mineral and biogenic components, Atmos. Chem. Phys., 14(4), 1853–1867, doi:10.5194/acp-14-1853-2014, 2014.

Peckhaus, A., Kiselev, A., Hiron, T., Ebert, M. and Leisner, T.: A comparative study of K-rich and Na/Ca-rich feldspar ice nucleating particles in a nanoliter droplet freezing assay, Atmos. Chem. Phys. Discuss., 0, 1–43, doi:10.5194/acp-2016-72, 2016.

Tobo, Y., Demott, P. J., Hill, T. C. J., Prenni, A. J., Swoboda-Colberg, N. G., Franc, G. D. and Kreidenweis, S. M.: Organic matter matters for ice nuclei of agricultural soil origin, Atmos. Chem. Phys., 14(16), 8521–8531, doi:10.5194/acp-14-8521-2014, 2014.

Vali, G., DeMott, P. J., M??hler, O. and Whale, T. F.: Technical Note: A proposal for ice nucleation terminology, Atmos. Chem. Phys., 15(18), 10263–10270, doi:10.5194/acp-15-10263-2015, 2015.